# Autoregressive Models for Small-Scale Knowledge Graph Generation

**Thiviyan Thanapalasingam**[*]                          *thiviyan.t@gmail.com*
*Universiteit van Amsterdam*
*1098XH Amsterdam, The Netherlands*

**Antonis Vozikis**[*]                          *a.vozikis@student.vu.nl*
*Vrije Universiteit Amsterdam*
*1081HV Amsterdam, The Netherlands*

**Peter Bloem**                          *p.bloem@vu.nl*
*Vrije Universiteit Amsterdam*
*1081HV Amsterdam, The Netherlands*

**Paul Groth**                          *p.t.groth@uva.nl*
*Universiteit van Amsterdam*
*1098XH Amsterdam, The Netherlands*

**Reviewed on OpenReview:** *https://openreview.net/forum?id=xhyOtB4uzb*

## Abstract

We study autoregressive sequence modeling for the generation of small Knowledge Graphs (KGs) over a shared set of entities and relations. Unlike link prediction, which scores triples independently, this task requires capturing interdependencies across multiple triples to produce graphs that exhibit the type, temporal, and connectivity regularities present in the training data. We present **ARK** (**A**uto-**R**egressive **K**nowledge Graph Generation), a family of autoregressive models that generate KGs by treating graphs as sequences of (head, relation, tail) triples. ARK learns these regularities directly from data, without explicit rule supervision and without being told the underlying constraints during training or inference. On the IntelliGraphs benchmark, our models achieve 89.2% to 100.0% semantic validity across its five datasets while generating novel graphs not seen during training. We also introduce **SAIL**, a variational extension of ARK that enables controlled generation through learned latent representations, supporting both unconditional sampling and conditional completion from partial graphs. We evaluate SAIL's controlled generation quantitatively through systematic conditioning experiments on wd-movies rather than relying solely on qualitative examples. Our analysis reveals that across four IntelliGraphs datasets, model capacity is more critical than architectural depth for this task, with recurrent architectures achieving comparable validity to transformer-based alternatives while offering substantial computational efficiency. These results establish autoregressive sequence models as strong baselines for the generation of small KGs over a shared vocabulary, providing a useful reference point for future work that may incorporate stronger inductive biases or broader empirical scope. Our code is available on `https://github.com/thiviyanT/ARK`.

## 1 Introduction

Knowledge Graphs (KGs) encode knowledge as graphs of entities connected by typed relations, powering applications from search engines to drug discovery (Hogan et al., 2021). However, even large-scale KGs

---

[*]Equal contribution.

such as Wikidata miss substantial world knowledge. Although Knowledge Graph Embedding (KGE) models address incompleteness (Bordes et al., 2013; Yang et al., 2015), they score each triple independently, failing to capture the interdependencies that define valid knowledge structures. This independence assumption becomes particularly problematic for complex facts requiring multiple related triples to represent accurately (Nathani et al., 2019).

In this work we study the generation of small Knowledge Graphs over a shared set of entities and relations, where the model learns from a training set of similar small graphs. Each graph is a small KG in its own right; one can view the union of all generated graphs as a large KG, but the task does not require this. Generation in this sense differs from link prediction (predicting individual triples), text-to-KG extraction (mapping text to KG facts), and conditional completion of a parent graph (sampling structures conditioned on an existing larger KG).

Generative modeling of complete graphs is a more general capability than these alternatives, rather than a niche alternative to them. A model that can sample $p(G)$ over complete graphs can, with minimal additional work, be used for tasks including link prediction, conditional completion, and structured imputation, in the same way that autoregressive language models trained for generation now serve as the foundation for a wide variety of downstream tasks. Settings where unconditional or joint multi-triple generation is genuinely preferable include sampling structures that satisfy joint regularities across multiple triples (e.g., $N$-ary temporal facts where start and end years must be consistent (Wen et al., 2016)), data augmentation requiring diverse novel graphs rather than completion of a specific input, and compression and density estimation over graph structures.

Consider representing the filmography of a director such as Tim Burton: a variable-size collection of films, cast members, and genres that co-vary jointly, where a triple-wise scorer can rank individual completions but cannot decide which set of films, actors, and genres belong together as a coherent whole. Generative modeling over complete graphs is a natural way to address this. Prior work in this direction includes generative reformulations of KGE models (Xiao et al., 2016; Cowen-Rivers et al., 2019; Loconte et al., 2024) and graph generative models from the molecular and structural graph generation literature (Vignac et al., 2023; Kong et al., 2023; Liu et al., 2024). These approaches differ from our setting in important ways, which we discuss in Section 5; in particular, they typically model per-triple distributions or assume unlabeled / molecular graph structures, rather than learning distributions over collections of typed-relation triples that exhibit joint regularities.

We observe that small KGs of this kind can be naturally represented as sequences of triples (head, relation, tail), suggesting that autoregressive sequence models may be well-suited for this task. We introduce **ARK** (**A**uto-**R**egressive **K**nowledge Graph Generation), a family of autoregressive models that generate KGs by sequentially predicting triples. We further present **SAIL** (**S**equential **A**uto-Regress**I**ve Knowledge Graph Generation with **L**atents), a probabilistic extension of ARK that enables controlled generation from learned latent distributions.

Crucially, in our setup the type, temporal, and connectivity regularities present in the data are used only as an evaluation metric, to test whether the model has captured them from the training distribution. They are not given to the model up front, neither during training nor during inference. On the IntelliGraphs benchmark (Thanapalasingam et al., 2023), ARK achieves 89.2% to 100.0% semantic validity across diverse datasets while producing novel graphs not seen during training. We emphasize a distinction that becomes important when interpreting our results: semantic validity measures satisfaction of learned regularities, not factual correctness. A generated graph can be semantically valid while still being factually incorrect, and we caution against over-interpreting validity numbers as evidence of factual reliability.

The empirical evidence supports the claim that autoregressive sequence models are strong baselines for the generation of small KGs over a shared vocabulary, as evaluated on the IntelliGraphs benchmark, rather than a broader claim about KG generation in general. Our contributions are as follows:

1. We introduce ARK, an autoregressive approach to small-scale KG generation that learns type, temporal, and connectivity regularities from data without explicit rule supervision, achieving 89.2% to 100.0% semantic validity on the IntelliGraphs benchmark;

2. We present SAIL, a variational extension that enables controlled generation through learned latent representations, supporting both unconditional sampling and conditional completion from partial graphs, evaluated quantitatively through systematic conditioning experiments on wd-movies;

3. We show that across the four IntelliGraphs datasets where we ran the capacity-vs-depth sweep, hidden dimensionality matters more than architectural depth, with single-layer GRUs matching deeper transformer performance while offering computational efficiency;

4. We release our models and code, establishing baselines for future work on KG generation. Our code is available at `https://github.com/thiviyanT/ARK`.

## 2 Preliminaries

**Knowledge Graph Generation**     We consider the task of generating small Knowledge Graphs $G = (E, R, T)$, where $E$ is a set of entities, $R$ is a set of relations, and $T \subseteq E \times R \times E$ is a set of triples drawn from a shared vocabulary fixed at training time. Given a training set $\mathcal{D} = \{G_1, \ldots, G_n\}$ of small KGs, the task is to learn a generative model $p_\theta(G)$ that produces new graphs $G' \sim p_\theta$ over the same vocabulary. This setting differs from link prediction (predicting individual triples), text-to-KG extraction (mapping text to KG facts), and conditional completion of a parent graph (sampling structures conditioned on a larger existing KG). Each $G_i \in \mathcal{D}$ is a small KG in its own right; one can view the union $\bigcup_i G_i$ as a large KG, but the task does not require this.

The training data exhibits regularities that the model must capture from data alone, without being given the regularities up front. These include, depending on the dataset, type regularities (e.g., only persons can occupy a director role), temporal regularities (e.g., start year precedes end year), and connectivity regularities (e.g., graphs forming valid path structures). These regularities are not enforced during training or generation: the model never sees them as explicit rules. Instead, they are used at evaluation time to assess whether the model has captured the structure of the training distribution. This is particularly relevant for $N$-ary relations and other complex facts that cannot decompose into independent binary predictions (Thanapalasingam et al., 2023; Wen et al., 2016).

**Definition 2.1** (Knowledge Graph Generation)**.** Given a training set of Knowledge Graphs $\mathcal{D} = \{G_1, \ldots, G_n\}$ over a shared vocabulary $\mathcal{V} = \mathcal{E} \cup \mathcal{R}$, learn a generative model $p_\theta(G)$ that can sample new graphs $G' \sim p_\theta$, where $G' \notin \mathcal{D}$. We use this model for: (1) unconditional sampling, (2) conditional completion from a partial graph, and (3) density estimation for compression and analysis.

**Definition 2.2** (Semantic Validity)**.** Let $\mathcal{S} = \{s_1, \ldots, s_k\}$ be a set of regularities present in the training data, where each $s_i$ is an algorithmically checkable property of a graph (e.g., type regularities, temporal consistency, connectivity). A generated graph $G$ is *semantically valid* if it satisfies all $s_i \in \mathcal{S}$. The set $\mathcal{S}$ is used only for evaluation; it is not provided to the model during training or inference. Concrete examples include: (1) start_year $\leq$ end_year (temporal consistency); (2) entity types matching relation requirements (e.g., only Person entities can fill the head position of `birthplace` relations); (3) graphs forming valid path structures (connected, acyclic, directional); and (4) directors being persons rather than organizations or genres. We note that semantic validity is distinct from factual correctness: a generated graph can satisfy all regularities in $\mathcal{S}$ while still being factually incorrect (e.g., a temporally consistent but historically wrong claim about who held an office). Validity numbers reported in this paper should be interpreted accordingly.

**Variational Inference**     To learn latent representations, we use the $\beta$-VAE framework (Kingma & Welling, 2013; Higgins et al., 2017), which aims to maximize the evidence lower bound (ELBO):

$$\mathcal{L}(\phi, \theta; G) = \mathbb{E}_{q_\phi(z|G)}[\log p_\theta(G|z)] - \beta \, \mathrm{KL}[q_\phi(z|G) \| p(z)]. \tag{1}$$

## 3 Sequential Decoding for Knowledge Graph Generation

We present our approach to Knowledge Graph generation through sequential decoding. We first describe how graphs are linearized into token sequences, then introduce ARK (**A**uto-**R**egressive **K**nowledge Graph

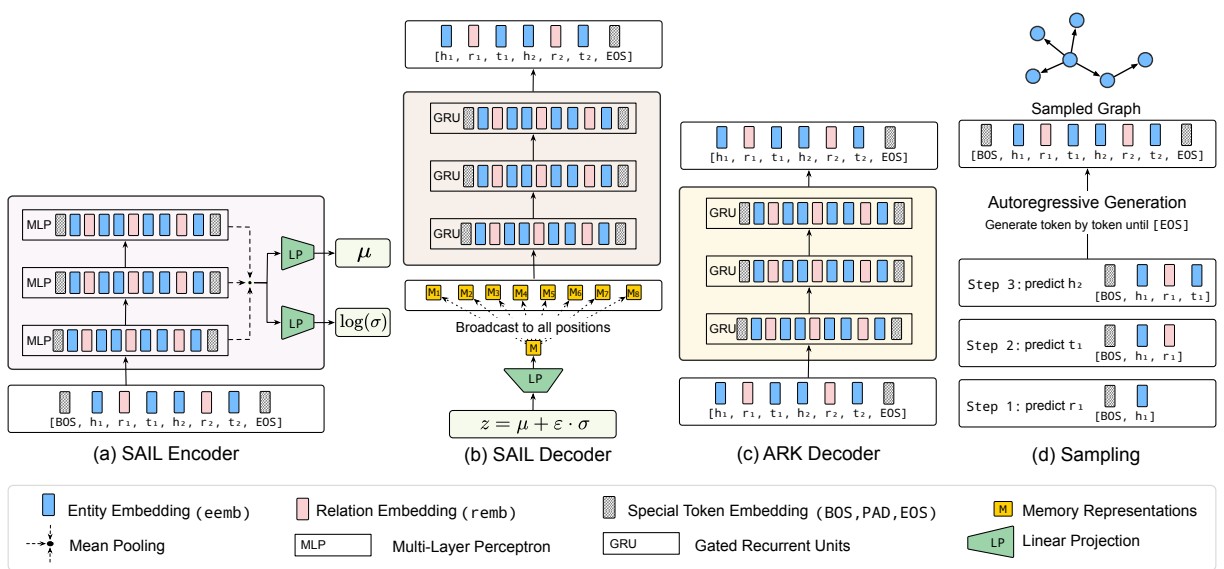

Figure 3.1: Overview of Model Architectures. **(a) SAIL Encoder:** Multi-layer perceptron (MLP) processes linearized KG sequences $[\texttt{BOS}, h_1, r_1, t_1, h_2, r_2, t_2, \ldots, \texttt{EOS}]$, with mean pooling to produce fixed-size representations. Linear projections generate latent distribution parameters $\mu$ and $\log \sigma$. **(b) SAIL Decoder:** GRU-based decoder conditions on sampled latent code $z \sim \mathcal{N}(\mu, \sigma^2)$ by broadcasting $z$ to all sequence positions and concatenating with embeddings $[M_1, M_2, \ldots, M_n]$ at each timestep. **(c) ARK Decoder:** GRU decoder for ARK operates without latent conditioning, processing embedded sequences directly through stacked GRU layers. **(d) Sampling:** Autoregressive generation proceeds token-by-token with causal masking until $\texttt{EOS}$ token or maximum length.

Generation), our autoregressive decoder model, followed by SAIL (**S**equential **A**uto-Regress**I**ve Knowledge Graph Generation with **L**atents), which extends ARK with a variational framework for controlled generation.

## 3.1 Graph Input Processing

To enable sequential generation, we linearize KGs into token sequences. A graph $G$ containing triples $(h_1, r_1, t_1), \ldots, (h_n, r_n, t_n)$ is represented as $[\texttt{BOS}, h_1, r_1, t_1, h_2, r_2, t_2, \ldots, h_n, r_n, t_n, \texttt{EOS}]$, where $\texttt{BOS}$ marks the sequence start and $\texttt{EOS}$ indicates termination. These tokens provide explicit generation boundaries, enabling the model to learn proper initiation and termination conditions. We employ a unified vocabulary $\mathcal{V} = \{\texttt{BOS}, \texttt{PAD}, \texttt{EOS}\} \cup \mathcal{E} \cup \mathcal{R}$ that combines special tokens, entities, and relations into a single embedding space. Variable-length graphs are padded to a fixed maximum length $L_{\max}$ using $\texttt{PAD}$ tokens for batched training.

During training, we randomize the order in which triples are presented within each graph, while preserving the (head, relation, tail) order within each triple, since that order carries semantic meaning (head and tail entities play asymmetric roles for non-symmetric relations). This is a deliberate design choice: a KG is naturally a set of triples rather than a sequence, and presenting triples in a fixed order at training time risks introducing leakage. The clearest example is syn-paths, in which each graph encodes a directed path. If triples were always presented in path order, the next-triple prediction problem would become trivial: the model could exploit the input ordering rather than learn the path structure of the graph. By shuffling the inter-triple order at training time, we force the model to learn a distribution over the set of triples rather than a particular linearization.

At evaluation time, we check whether the set of generated triples satisfies the dataset regularities, independent of the order in which they were produced. The model is therefore trained to be approximately invariant to inter-triple ordering and evaluated on the resulting set of triples. We confirm empirically that the choice of training-time inter-triple ordering does not drive results: training under shuffled, alphabetical, and dataset-specific orderings gives equivalent semantic validity, novelty, and compression scores (Appendix A.9).

### 3.2 Autoregressive Knowledge generation (ARK)

ARK is an autoregressive model that generates KGs token-by-token, predicting each element conditioned on all previous tokens. We use Gated Recurrent Units (GRUs) (Cho et al., 2014) as the sequence model, exploiting the natural sequential structure of linearized graphs. See Appendix A.3.1 for architectural details.

The model is trained autoregressively with cross-entropy loss, conditioning on ground-truth previous tokens: $\mathcal{L}_{\text{ARK}} = -\sum_{t=1}^{T} \log p(x_t|x_{<t})$.

**Generation**   During inference, ARK generates graphs sequentially starting from the BOS token. At each timestep $t$, the model computes the probability distribution $p(x_{t+1}|x_{\leq t})$ over the vocabulary. We select the next token through sampling controlled by temperature and top-$k$. Concretely, we divide logits by temperature $T$, keep only the top-k tokens, then retain the smallest prefix whose cumulative probability mass exceeds $p$ (top-$p$), renormalize and sample one token. Decoding stops on EOS or when the maximum graph length has been reached. The generated sequence is parsed into triples by extracting consecutive $(h, r, t)$ token triplets between BOS and EOS markers, with incomplete triples discarded during post-processing.

### 3.3 Sequential Autoregressive Knowledge Graph Generation with Latents (SAIL)

SAIL extends ARK by incorporating a variational autoencoder framework, similar to Bowman et al. (2016), enabling probabilistic generation from learned latent distributions, $z$. This extension allows for controlled generation and interpolation in latent space while maintaining the efficiency of GRU-based decoding. The VAE framework enables: (1) controlled generation through latent manipulation, (2) interpolation between graphs, and (3) conditional generation from partial graphs. The technical challenge is learning meaningful, continuous representations of discrete graph structures.

**Encoder**   The encoder treats the input as a set of triples and produces a fixed-size graph-level representation. Each triple $(h, r, t)$ is embedded as $[E_e[h]; E_r[r]; E_e[t]] \in \mathbb{R}^{3d}$, where the order within the embedding preserves the asymmetric roles of head, relation, and tail. We then mean-pool across triples to obtain a graph-level vector that is invariant to the inter-triple order, matching the order-randomization used at training time. The resulting vector is passed through a multi-layer perceptron (MLP) whose number of dense layers matches the number of stacked GRU layers in the decoder, with GELU activations. The final hidden representation is projected to latent distribution parameters, $\boldsymbol{\mu}$ and $\log \boldsymbol{\sigma}^2$.[1]

**Latent Sampling**   We sample from the latent distribution using the reparameterization trick:

$$\mathbf{z} = \boldsymbol{\mu} + \boldsymbol{\sigma} \odot \boldsymbol{\epsilon}, \quad \boldsymbol{\epsilon} \sim \mathcal{N}(0, \mathbf{I}) \tag{2}$$

Following standard VAE practice, we use a fixed prior $p(z) = \mathcal{N}(0, I)$ rather than learning it, which acts as a regularizer. The learned components of our model are the encoder $q_\phi(\mathbf{z}|G)$, which maps graphs to latent distributions, and the decoder $p_\theta(G|\mathbf{z})$, which reconstructs graphs from latent codes. Despite using this simple fixed prior, our t-SNE visualizations in Figure 4.1 demonstrate that the learned posterior captures meaningful structure, with clear clustering by genre.

**Decoder**   The decoder extends ARK's GRU architecture by conditioning on the latent variables, $\mathbf{z}$. The latent representation is first projected and used to initialize the decoder's hidden state: $\mathbf{h}_0 = \tanh(\mathbf{W}_{\text{init}}\mathbf{z} + \mathbf{b}_{\text{init}})$ To maintain global conditioning throughout generations, $\mathbf{z}$ is broadcast to all sequence positions. At each timestep, we concatenate the projected latent code with the input embedding: $\mathbf{x}'_t = [\mathbf{x}_t; \mathbf{W}_z\mathbf{z}]$ This ensures that the global graph structure encoded in $\mathbf{z}$ influences every token prediction, allowing the decoder to maintain semantic consistency across the entire sequence. SAIL is trained by maximizing the ELBO (as shown in Equation 1).

---

[1]It may seem counter-intuitive to use a set-pooling MLP encoder rather than a sequential encoder. In preliminary experiments, a GRU-based encoder appeared to perform notably worse than the MLP encoder. To understand this, we ran a diagnostic comparison between the MLP encoder, a GRU encoder, and a small transformer encoder, reported in Appendix A.10. After correcting an evaluation issue in the GRU generation path, the three encoders produce comparable downstream validity, and the GRU encoder shows no signs of posterior collapse. We retain the MLP encoder for SAIL because it is the simplest of the three and matches the inter-triple permutation invariance built into the training procedure.

**Generation & Sampling**  To generate a graph using the model, we sample $\mathbf{z} \sim \mathcal{N}(0, \mathbf{I})$ from the prior distribution. We call this *unconditional generation*. Additionally, we define *conditional generation* where we encode a partial graph to obtain the posterior $q(\mathbf{z}|G_{\text{partial}})$, sample from it, and then complete the sequence. The generation then follows an autoregressive process where the probability of the complete graph factorizes as: $p_\theta(G|\mathbf{z}) = \prod_{t=1}^{T} p_\theta(x_t|x_{<t}, \mathbf{z})$. We use beam search with $\text{score}(x_{1:t}|\mathbf{z}) = \sum_{i=1}^{t} \log p_\theta(x_i|x_{<i}, \mathbf{z})$. Latent conditioning enables controlled generation by manipulating $\mathbf{z}$, we can interpolate between graphs or explore specific regions of the latent space to generate graphs with desired properties.

## 4 Evaluation

We evaluate ARK, SAIL, and a family of comparison models on the IntelliGraphs benchmark (Thanapalasingam et al., 2023), which consists of five datasets designed to test different aspects of small-KG generation.

**Benchmark**  IntelliGraphs includes three synthetic datasets (syn-paths, syn-types, syn-tipr) with algorithmically verifiable semantics, ranging from simple path structures to temporal constraints requiring reasoning about time intervals, and two real-world Wikidata-derived datasets (wd-movies, wd-articles) capturing complex relational patterns from movie and academic publication domains. Synthetic datasets contain fixed-size graphs (3-5 triples) with small vocabularies (30-130 entities), while Wikidata datasets feature variable-size graphs (2-212 triples) with large entity vocabularies (24K-61K entities), providing diverse challenges for evaluating generation quality and semantic validity. Detailed dataset characteristics and semantic constraints are provided in Appendix A.2. IntelliGraphs is, to our knowledge, the benchmark most directly designed for the generation of small KGs over a shared vocabulary; other KG benchmarks focus on link prediction or text-to-KG extraction. Each instance in IntelliGraphs is itself a small KG, generated and evaluated as a complete graph rather than as a substructure conditioned on a parent graph.

**Baselines**  We compare ARK and SAIL against three families of baselines: the original IntelliGraphs benchmark baselines, transformer-based variants of our own architectures, and additional generative model families to control for the choice of model class.

*IntelliGraphs probabilistic baselines.* The probabilistic baselines from Thanapalasingam et al. (2023) decompose graph generation as $p(F) = p(S \mid E)p(E)$, where $E$ represents entities and $S$ represents structure. The *uniform* baseline samples from uniform distributions, providing reference compression bits under the assumption of equal likelihood for all configurations. The KGE-based baselines (TransE, ComplEx, DistMult) estimate $p(E)$ using entity frequencies with Laplace smoothing and $p(S \mid E)$ using learned scoring functions: TransE models relations as translations (Bordes et al., 2013), DistMult uses bilinear interactions (Yang et al., 2015), and ComplEx employs complex-valued embeddings (Trouillon et al., 2016). These baselines were designed for triple-level scoring rather than for joint graph generation; we include them because they are the established comparison set for IntelliGraphs and because their behavior is informative about what a per-triple approach produces when asked to generate complete graphs.

*Transformer-based variants.* To study the contribution of the recurrent decoder, we implement transformer-based variants of our own models: $t$-ARK uses a transformer decoder with causal self-attention, while $t$-SAIL extends this with a variational framework employing transformer encoders and decoders. These variants allow us to examine whether attention mechanisms provide benefits over GRU-based decoding for this task.

*Cross-family generative baselines.* To control for the choice of model class (autoregressive vs. latent-variable vs. diffusion) and to test whether the autoregressive formulation specifically drives strong performance, we additionally compare against a plain VAE and a diffusion-based model trained under identical conditions to ARK and SAIL on all five IntelliGraphs datasets. The plain VAE encodes the full triple-set into a single latent and decodes it to a graph in one shot (no autoregressive decoding); the diffusion model iteratively denoises a noise-corrupted graph representation. Both use the same vocabulary, the same input encoding, the same hyperparameter search budget, and the same evaluation protocol as our other models. We report these as Plain-VAE and Diffusion in Table 1.

*Methods considered but not run as direct baselines.* Several adjacent methods were considered as potential external baselines, including KGT5 (Kochsiek et al., 2023), the circuit-based generative KGE of Loconte et al. (2024), GraphARM (Kong et al., 2023), GraphMaker (Liu et al., 2024), and DiGress (Vignac et al., 2023). Each requires non-trivial adaptation to fit the IntelliGraphs setting: KGT5 is designed for conditional triple prediction, not joint graph generation, and an adapted version reduces to our $t$-ARK variant; Loconte et al. (2024) defines per-triple distributions, and joint-graph sampling would require constructing a multi-triple distribution from per-triple scores; GraphARM, GraphMaker, and DiGress assume node feature vectors and small categorical edge label sets, which do not transfer cleanly to KGs with relation-typed edges and entity vocabularies of 24K-61K. In each case the adaptation would substantially shape the result, so any reported number would primarily reflect the quality of our adaptation rather than the underlying method. See Section 5 for further discussion.

**Evaluation Metrics**    We evaluate generation quality through three primary metrics: (1) *Semantic Validity* – the proportion of generated graphs that satisfy the regularities present in the training data, measured by an algorithmic checker (see Appendix A.2); (2) *Novelty* – the proportion of generated graphs not present in the training set, distinguishing genuine generation from memorization; and (3) *Compression* – the information-theoretic measure $-\log p(G)$ in bits, quantifying how efficiently the model encodes graph structure. For variational models, we additionally report the KL divergence between the approximate posterior and prior. Semantic validity measures satisfaction of learned regularities, not factual correctness. A generated graph can be semantically valid (e.g., satisfy all type and temporal regularities of wd-movies) while still being factually incorrect (e.g., attribute a film to the wrong director). The validity numbers reported below should not be interpreted as evidence of factual reliability.

## 4.1   Compression Code Length

We express the negative log-likelihood, $-\log_2(p_\theta)$, in bits-per-graph. See Appendix A.3.2 for details. This measures both the ability to compress and to predict (Grünwald, 2007, Section 3.2).

**Results**    Table 1 shows the compression performance across all models. ARK achieves strong compression rates across all datasets, with 27.65 bits for syn-paths (compared to 30.49 bits for the uniform baseline) and 23.48 bits for syn-tipr. On real-world datasets, ARK achieves the best overall compression with 98.19 bits for wd-movies and 205.24 bits for wd-articles, demonstrating efficient encoding of complex graph structures. While their compression on syn-types is higher (59.63 and 59.79 bits), both models compensate with strong semantic validity in generation tasks. By contrast, the variational models (SAIL and $t$-SAIL) report ELBO upper bounds rather than exact compression, as they use latent vectors **z** to capture graph structure. Their compression includes both reconstruction and KL divergence terms, with the KL component varying from nearly zero to syn-types (0.15 bits) to moderate values on other datasets (13-32 bits), indicating adaptive latent space usage on dataset complexity.

## 4.2   Sampling from Latent Variable, $z$

We assess the generative capabilities of SAIL through two complementary approaches: unconditional generation by sampling from the prior distribution $p_\theta(z)$, and conditional generation by providing partial graph sequences. These experiments test whether the learned latent space is well-structured and whether the model can generate semantically valid, novel graphs, demonstrating true generative modeling rather than mere memorization. For more details regarding the method and qualitative analysis, we refer the reader to Appendices A.3.3 and A.4, respectively.

**Quantitative Results**    Table 1 shows unconditional graph generation results. ARK achieves high semantic validity across synthetic datasets: 99.95% on syn-paths, 100.00% on syn-tipr, and 89.22% on syn-types. SAIL demonstrates similarly strong performance with 92.50%, 98.45%, and 100.00% validity, respectively. Both models substantially outperform the original IntelliGraphs KGE baselines (TransE, DistMult, ComplEx), which achieve less than 1% validity and produce 76-100% empty graphs. This result is expected given that those baselines were designed to score individual triples rather than to generate joint graph structures, and we report it primarily to establish the comparison with the original benchmark. All generated graphs from our models are novel rather than memorizing training examples. For real-world datasets, ARK main-

| Datasets | Model | % Valid Graphs ↑ | % Novel & Valid ↑ | % Empty Graphs ↓ | Compression Length (bits/graph) ↓ |
|---|---|---|---|---|---|
| **syn-paths** | uniform | 0 | 0 | 0 | 30.49 |
| | TransE | 0.25 | 0.25 | 76.55 | 49.89 |
| | DistMult | 0.69 | 0.69 | 85.41 | 54.39 |
| | ComplEx | 0.71 | 0.71 | 85.73 | 48.58 |
| | VAE | 76.21 | 76.21 | 0 | 41.33 |
| | Diffusion | 52.44 | 52.44 | 0 | 43.27 |
| | $t$-SAIL | 99.60 | 99.60 | 0 | 27.77 |
| | SAIL | 92.50 | 92.50 | 0 | 28.74 |
| | $t$-ARK | 97.39 | 97.39 | 0 | **27.57** |
| | ARK | **99.95** | **99.95** | 0 | 27.65 |
| **syn-tipr** | uniform | 0 | 0 | 0 | 61.61 |
| | TransE | 0 | 0 | 94.42 | 69.51 |
| | DistMult | 0 | 0 | 86.66 | 63.96 |
| | ComplEx | 0 | 0 | 96.05 | 67.51 |
| | VAE | 0 | 0 | 0 | 18.74 |
| | Diffusion | 0 | 0 | 0 | 18.96 |
| | $t$-SAIL | 100.00 | 100.00 | 0 | 26.30 |
| | SAIL | 98.45 | 98.45 | 0 | 27.14 |
| | $t$-ARK | 100.00 | 100.00 | 0 | **23.34** |
| | ARK | **100.00** | **100.00** | 0 | 23.48 |
| **syn-types** | uniform | 0 | 0 | 0 | **36.02** |
| | TransE | 0.21 | 0.21 | 84.56 | 48.26 |
| | DistMult | 0.13 | 0.13 | 87.53 | 47.46 |
| | ComplEx | 0.07 | 0.07 | 89.75 | 47.69 |
| | VAE | 0 | 0 | 0 | 49.74 |
| | Diffusion | 0 | 0 | 0 | 46.31 |
| | $t$-SAIL | **100.00** | **100.00** | 0 | 59.61 |
| | SAIL | 100.00 | 100.00 | 0 | 60.58 |
| | $t$-ARK | 87.07 | 87.07 | 0 | 59.79 |
| | ARK | 89.22 | 89.22 | 0 | 59.63 |
| **wd-movies** | uniform | 0 | 0 | 0 | 171.60 |
| | TransE | 0 | 0 | 85.39 | 208.60 |
| | DistMult | 0 | 0 | 87.07 | 202.68 |
| | ComplEx | 0 | 0 | 98.13 | 208.50 |
| | VAE | 0 | 0 | 0 | 148.32 |
| | Diffusion | 0 | 0 | 0 | 154.17 |
| | $t$-SAIL | **99.83** | **99.83** | 0 | 124.50 |
| | SAIL | 99.47 | 99.47 | 0 | 116.84 |
| | $t$-ARK | 98.33 | 98.33 | 0 | 114.49 |
| | ARK | 99.24 | 99.24 | 0 | **98.19** |
| **wd-articles** | uniform | 0 | 0 | 0 | 693.80 |
| | TransE | 0 | 0 | 95.42 | 910.65 |
| | DistMult | 0 | 0 | 100.00 | 887.30 |
| | ComplEx | 0 | 0 | 97.54 | 901.91 |
| | VAE | 0 | 0 | 0 | 205.68 |
| | Diffusion | 0 | 0 | 0 | 219.74 |
| | $t$-SAIL | 98.00 | 98.00 | 0 | 235.24 |
| | SAIL | **99.13** | **99.13** | 0 | **199.55** |
| | $t$-ARK | 95.37 | 95.37 | 0 | 224.25 |
| | ARK | 97.24 | 97.24 | 0 | 205.24 |

Table 1: Semantic validity and compression length in bits of the graphs generated. We sample graphs and check the novelty of the sampled graphs by comparing them against the training and validation sets. We use the test set for the calculation of the compression length when training has finished. The best performing models for each dataset are **bolded**. Baseline results are from the IntelliGraphs paper (Thanapalasingam et al., 2023). The full results are available in Tables 5 and 6 in the Appendix.

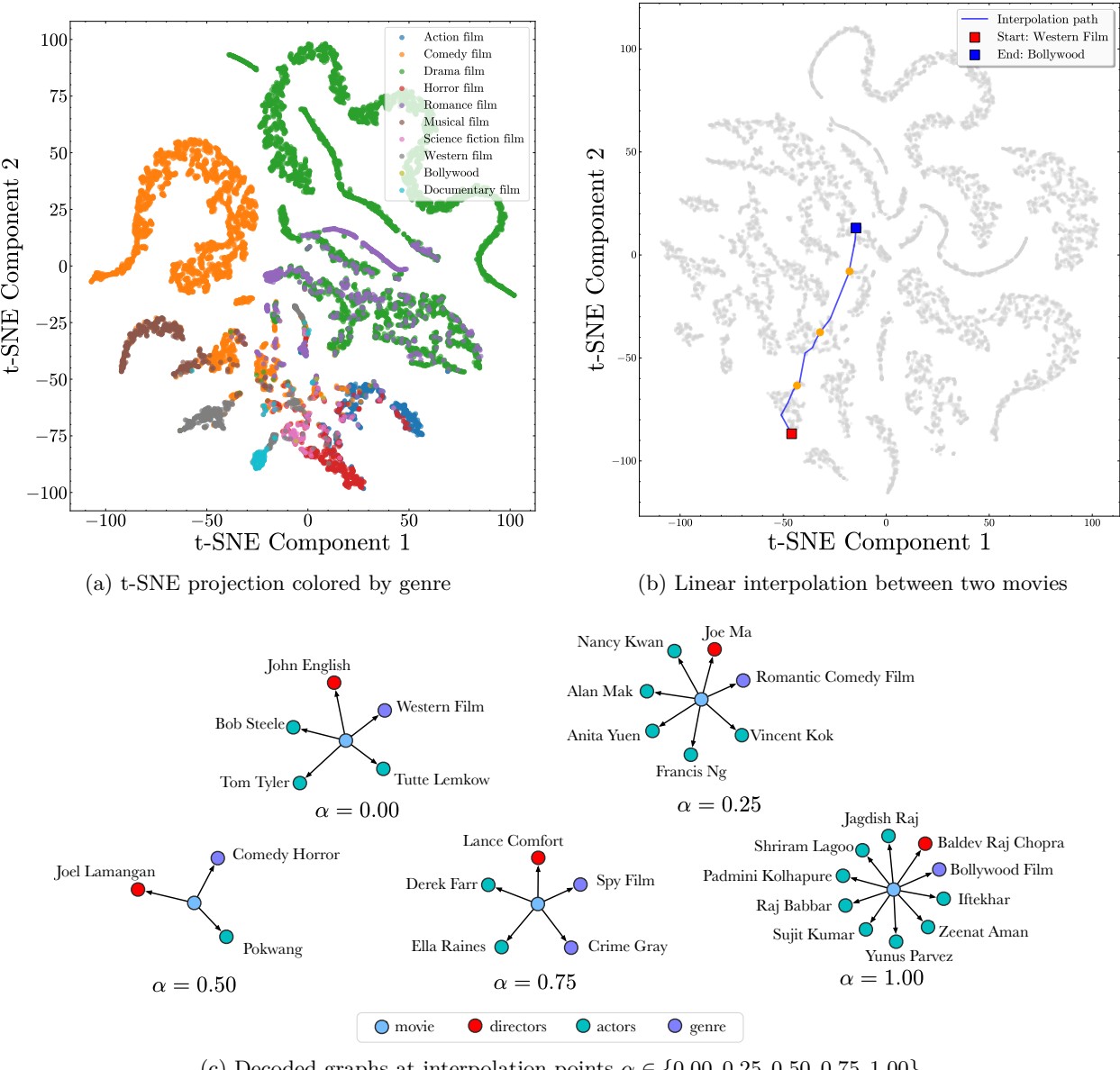

(a) t-SNE projection colored by genre

(b) Linear interpolation between two movies

(c) Decoded graphs at interpolation points $\alpha \in \{0.00, 0.25, 0.50, 0.75, 1.00\}$

Figure 4.1: Latent space visualization for the wd-movies dataset. (a) t-SNE projection shows clear clustering by genre. (b) Smooth interpolation paths connect different movie types. (c) Decoded graphs along the interpolation path show gradual transitions in cast and genre attributes, maintaining semantic validity throughout.

tains 99.24% validity on wd-movies and 97.24% on wd-articles, while SAIL achieves 99.47% and 99.13% respectively, demonstrating robust performance despite increased complexity.

The cross-family baselines (VAE and Diffusion) provide a more informative comparison. The plain VAE achieves 76.21% validity on syn-paths and 0% on wd-movies, while the diffusion-based model achieves 52.44% and 0%. On wd-movies, both models also fail to produce any novel and valid graphs, yielding 0% on this combined metric, which highlights their inability to generalize to more complex real-world structures.

### 4.3 Interpolation in Latent Space

For SAIL and $t$-SAIL, we explore the structure of the learned latent space by interpolating between encoded representations of different graphs. This analysis reveals whether the model learns smooth, semantically meaningful transitions between graph structures, indicating a well-organized latent space where similar graphs cluster together and intermediate points correspond to valid hybrid structures. For more details regarding the method, we refer the reader to Appendix A.3.4.

**Quantitative Results** The smoothness metrics reveal distinct patterns across dataset complexity. Comparing SAIL and $t$-SAIL (Table 7 in Appendix), we observe that $t$-SAIL generally achieves better latent space organization. For syn-tipr, $t$-SAIL shows exceptional quality with near-perfect local smoothness (0.99) and global consistency (0.98), while SAIL achieves lower but still strong metrics (0.93 and 0.69, respectively). The architectural difference is most pronounced on syn-paths, where $t$-SAIL maintains moderate global consistency (0.36) compared to SAIL's much weaker performance (0.14), suggesting that transformer-based encoders better capture graph structure. Surprisingly, SAIL demonstrates superior performance on syn-types with high local smoothness (0.92) and global consistency (0.73), exceeding $t$-SAIL's metrics (0.82 and 0.60). The flip rates reveal interesting trade-offs: SAIL shows higher instability on most datasets (0.33 for syn-paths, 0.40 for wd-movies) compared to $t$-SAIL (0.20 and 0.15 respectively), though they achieve similar rates on syn-tipr (0.10 vs. 0.09). Real-world datasets (wd-movies) show both models achieving strong local smoothness (0.84 and 0.87) but with $t$-SAIL maintaining better global consistency (0.58 vs 0.49). These results suggest that while transformer encoders generally provide better latent space organization, the simpler GRU-based SAIL can match or exceed transformer performance on certain structured datasets, particularly syn-types.

**Qualitative Results** Figure 4.1 demonstrates the learned latent space structure for the wd-movies dataset. The t-SNE projection (Figure 4.1 a) reveals distinct clustering by genre, indicating that SAIL learns to organize its latent space according to semantic film categories without explicit supervision. The linear interpolation experiment (Figure 4.1 b,c) traces a path between a Western film and a Romantic Comedy, with decoded graphs at intermediate points ($\alpha \in \{0, 0.25, 0.5, 0.75, 1\}$) demonstrating smooth transitions: starting from a Western with actors Bob Steele and Tom Tyler, progressing through hybrid representations with mixed genre elements (Comedy Horror at $\alpha = 0.50$), and reaching a Thriller film with different cast members. While all intermediate graphs maintain a valid KG structure, the semantic coherence varies; intermediate points produce valid but potentially less realistic combinations of actors and genres, suggesting that semantic validity is preserved throughout interpolation, but semantic plausibility is highest near the training data manifold, consistent with typical VAE behavior on structured data.

### 4.4 Ablation Study

The ablation study in Appendix A.6 examines the relationship between model capacity and generation quality on syn-paths, with extensions to three additional datasets in Appendix A.7. We find that hidden dimensionality is more critical than network depth, with a dataset-dependent saturation point that holds consistently across the four datasets we swept. Single-layer GRU models with sufficient capacity match the validity of deeper architectures on every dataset, suggesting that the depth-insensitivity finding is not specific to syn-paths but generalizes within the IntelliGraphs benchmark. Compression efficiency improves with model complexity, but lightweight architectures match transformer-based models in generation quality.

## 5 Related Work

We position our work in the context of four lines of related research: generative modeling of Knowledge Graphs, sequence-based KG modeling, reasoning over Knowledge Graphs, and general graph generative models. For each, we describe the points of contact with our setup and the points of difference, so that readers can locate where ARK and SAIL sit in this landscape.

**Generative Modeling of Knowledge Graphs** Several lines of work have studied generative modeling of KG facts, although their target tasks differ from ours. Cowen-Rivers et al. (2019) learn joint probability

distributions over facts in a KG to estimate the predictive uncertainty of KGE models, and evaluate using link prediction rather than graph generation. Xiao et al. (2016) (TransG) is a probabilistic model that learns the semantics of $N$-ary relations through a Bayesian non-parametric mixture, but again models per-triple rather than per-graph distributions. Loconte et al. (2024) reinterpret the score functions of traditional KGEs as probabilistic circuits, enabling efficient marginalization and sampling and thereby providing a principled way to sample new triples consistent with an existing KG. All three of these works share with our approach the use of probabilistic modeling over KG facts, but each defines distributions over individual triples rather than over collections of triples that exhibit joint regularities. Adapting them to sample complete graphs evaluable under joint regularities would require non-trivial extensions, which we discuss further in Section 4.

**Sequence-Based KG Modeling**   A separate line of work uses sequence models over linearized KG facts. KGT5 (Kochsiek et al., 2023) trains a T5-style seq2seq model on triple-level inputs for KG completion and KG-question answering, treating completion as conditional generation of a missing entity given a partial triple. KGGLM (Balloccu et al., 2024) similarly uses generative language modeling for KG representation learning in the context of recommender systems. We share with these approaches the design choice of representing KG facts as token sequences and training autoregressive models over them. We differ in the target task: KGT5 and KGGLM model conditional distributions at the triple level (completing or scoring individual facts), whereas we model joint distributions over collections of triples. Therefore there is an architectural overlap, but the research question being asked is different. To the extent that an adapted version of KGT5 produces complete graphs by chaining triple-level completions, the resulting model is functionally close to our $t$-ARK variant.

**Reasoning over Knowledge Graphs**   Several methods score, retrieve, or reason over structures within an existing KG. Neelakantan et al. (2015) use RNNs to score paths for link prediction; Zhang et al. (2024) address triple set prediction in a closed-world setting; Sun et al. (2018) focus on entity alignment between KGs. Galkin et al. (2024) (ULTRA) propose a foundation model for KG reasoning that generalizes across diverse KGs through pre-training on multiple graphs. These approaches share with our work an interest in modeling structure beyond individual triples. They differ in that they operate over a given parent graph, scoring or retrieving substructures, rather than sampling new graphs from a learned distribution. The goal in this line of work is conditional reasoning over an existing graph; the goal in our work is unconditional or partially-conditioned generation.

**Graph Generative Models**   A substantial body of work studies generative modeling of graphs in the molecular and structural graph generation literature. GraphVAE and GraphRNN model graph distributions through latent variables and autoregressive decoding respectively (Li et al., 2018). Kipf et al. (2020) infer relational structure from observations and visual data. More recent work includes DiGress (Vignac et al., 2023), which applies discrete denoising diffusion to graph generation; GraphARM (Kong et al., 2023), which combines autoregressive models with graph neural networks; and GraphMaker (Liu et al., 2024), a diffusion-based approach that iteratively refines node features and edge structures. These methods share with our work the goal of generating novel graphs from a learned distribution. They differ in their assumptions about the graph structure: most assume node feature vectors and small categorical edge label sets, which are appropriate for molecular graphs but do not transfer cleanly to KGs with relation-typed edges and large entity vocabularies (24K–61K unique entities in the Wikidata-derived datasets). Adapting these methods to the IntelliGraphs setting would require either embedding entities into a feature space or modifying the architecture to handle categorical entities natively, both of which substantially affect the resulting comparison. We therefore do not run these methods as direct baselines and instead provide a controlled cross-family comparison using a plain VAE and a diffusion-based model trained under identical conditions to ARK and SAIL (see Section 4). Architectures originally proposed for encoding existing graphs, such as GraphGPS (Rampášek et al., 2022), NodeFormer (Wu et al., 2022), and Exphormer (Shirzad et al., 2023), are complementary to this work: they address the encoding side, whereas we address the generation side, and combining them is a natural future direction.

**Neuro-Symbolic Generative Models for KGs**   Combining distributed and symbolic representations, neuro-symbolic systems aim to combine the strengths of both paradigms (van Bekkum et al., 2021). Jiang & Ahn (2020) combine distributed and symbolic entity-based representations in a generative latent variable model to infer object-centric symbolic representations from images. van Krieken et al. (2025) introduce a

neurosymbolic diffusion model that integrates logical constraints directly into the diffusion process. We share with these approaches the goal of producing structured outputs that respect logical or relational regularities. We differ in that we do not enforce symbolic constraints during generation: in our setup, the regularities present in the data are learned implicitly from training examples and used only at evaluation time. This is a deliberate design choice rather than a limitation; whether explicit constraint integration would improve performance on IntelliGraphs is an open question we leave to future work.

## 6  Conclusion

We have studied autoregressive sequence modeling as an approach to the generation of small Knowledge Graphs over a shared set of entities and relations. By representing each graph as a sequence of (head, relation, tail) triples, our models learn the type, temporal, and connectivity regularities present in the training data without explicit rule supervision. On the IntelliGraphs benchmark, ARK and SAIL achieve 89.2% to 100.0% semantic validity across all five datasets, substantially outperforming KGE baselines that score triples independently. The variational extension SAIL enables controlled generation through learned latent representations, supporting both unconditional sampling and conditional completion from partial graphs.

Our analysis reveals that across the four IntelliGraphs datasets where we ran the capacity-vs-depth sweep (syn-paths, syn-types, syn-tipr, wd-movies), hidden dimensionality matters more than architectural depth, with single-layer GRU models matching deeper transformer performance while offering substantial computational efficiency. We further find that a plain VAE and a diffusion-based model trained under identical conditions do not outperform the autoregressive formulation in our experiments. On this benchmark, the autoregressive sequential decoder appears to matter more than the choice of generative framework; further studies are needed to determine whether this finding generalizes to larger graphs or other settings.

Autoregressive sequence models are strong baselines for the generation of small KGs over a shared vocabulary, as evaluated on IntelliGraphs. Whether the same architectures and findings transfer to larger graphs, open-world vocabularies, or settings outside this benchmark is an open empirical question that we leave to future work. One further caveat is worth restating: semantic validity measures satisfaction of regularities learned from data, not factual correctness. A generated graph can be semantically valid while still being factually wrong, and our high validity numbers should not be read as evidence of factual reliability.

**Limitations** Our setting assumes a fixed vocabulary of entities and relations known at training time, and the graphs in our experiments contain 3 to 212 triples. The setting therefore does not address open-world scenarios where new entities emerge dynamically, nor does it test scaling to substantially larger graphs. The autoregressive formulation imposes a linear ordering on inherently unordered graph structures; our experiments comparing shuffled, alphabetical, and dataset-specific orderings show that this does not affect generation quality on IntelliGraphs, but the question is worth revisiting at larger scales. The empirical scope is also limited to a single benchmark, and broader claims about KG generation in general are not supported by our experiments alone.

**Future Work** Several directions follow from this work. First, extending the framework to handle out-of-vocabulary entities and relations through compositional embeddings or meta-learning would relax the fixed-vocabulary assumption. Second, scaling beyond 200–500 triples through hierarchical generation strategies (generating local structures and composing them) would test the approach on larger graphs. Third, integrating stronger graph-aware inductive biases (GNN-based encoders, structural attention, or hybrid architectures) with autoregressive decoding is a non-trivial but worthwhile direction, since the cross-family comparison reported in Section 4 suggests that the choice of decoder matters more than the choice of generative framework on this benchmark. Fourth, integration with LLMs to combine structured generation with natural language understanding is a natural next step. Finally, larger and more complex KG generation benchmarks are needed: our models solve most challenges in IntelliGraphs, and the field would benefit from harder evaluation settings.

**Ethics Statement**    Datasets on which our models are trained may contain societal biases and factual errors, which could propagate through the learning process and manifest in generated knowledge graphs. While our models achieve high semantic validity scores, they may still reproduce or amplify biases present in the training data, potentially generating graphs that reflect historical inequities or stereotypes. Additionally, the autoregressive generation process could produce factually incorrect but semantically valid triples, as the model learns logical rules rather than verifying the truth. We intend for ARK and SAIL to be treated as research prototypes to advance the field of KG generation, and should not be deployed in critical applications without thorough testing and safeguards. See Thanapalasingam et al. (2023) for a detailed analysis of the limitations of the datasets.

**Reproducibility Statement**    We provide complete code and detailed configurations to ensure complete reproducibility of all experiments. Our implementation, including model architectures, training scripts, data preprocessing pipelines, and evaluation metrics, is available at `https://github.com/thiviyanT/ARK`. Experimental details, including hyperparameters, hardware specifications, and training procedures, are provided in Appendix A.1. We also release pre-trained model checkpoints for both ARK and SAIL to facilitate reproduction of our results and enable further research building upon our work. Detailed instructions for replicating each experiment, including expected runtimes and resource requirements, are provided in the repository.

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

# A Appendix

## A.1 Experimental Details

We used the PyTorch library [2] to develop and test the models. All experiments were performed on a single-node machine with an Intel(R) Xeon(R) Gold 5118 (2.30GHz, 12 cores) CPU and 64GB of RAM, with a single NVIDIA A100 GPU (80GB of VRAM) or a single NVIDIA H100 GPU (80GB of VRAM). We used PyTorch's CUDA acceleration for model training and inference. We used the Adam optimizer with variable learning rates (Kingma & Ba, 2015). We monitored the training of the models using the Weights & Biases package [3]. All experiments use the same train/validation/test splits as the original IntelliGraphs benchmark (Thanapalasingam et al., 2023) to ensure fair comparison.

**Hyperparameter Optimization** For ARK and SAIL, the hyperparameters were automatically tuned using grid search {learning rate, batch size, number of epochs, latent dimension size [4], number of neurons and number of layers[5] } to get the best performance for the validation split. For reproducibility, we provide an extension description of the hyperparameters as YAML files under the *configs* directory on `https://github.com/thiviyanT/ARK`.

## A.2 Dataset Details

The IntelliGraphs benchmark datasets test different aspects of semantic validity and structural complexity:

1. **syn-paths:** A synthetic dataset containing path graphs with simple semantics that can be algorithmically verified in linear time. These are acyclic graphs where edge directions follow the path structure.

2. **syn-types:** A synthetic dataset featuring typed entities and relations where type constraints on entities depend on the relation type, enforcing semantic consistency through type checking.

3. **syn-tipr:** A synthetic dataset containing subgraphs based on the *Time-indexed Person Role* (tipr) ontology pattern.[6] The semantics are defined by the tipr graph pattern, requiring temporal reasoning to generate valid time intervals.

4. **wd-movies:** Small knowledge graphs describing movies, extracted from Wikidata.[7] Each graph contains one existential node representing the movie, with entity nodes for director(s) connected via `has_director`, cast members connected via `has_actor`, and genres connected via `has_genre` relations.

5. **wd-articles:** Small knowledge graphs that describe research articles, extracted from Wikidata. Each graph contains one existential node representing the article, with entity nodes for author(s) connected via `has_author`, publication venues connected via `published_in`, and topics connected via `has_topic` relations.

---

[2] `https://pytorch.org/`

[3] `https://wandb.ai`

[4] for SAIL only

[5] Both encoder's and decoder's neurons and number of layers. For models without encoder the tuning for the number of layers and neurons was done the decoder part

[6] http://ontologydesignpatterns.org/wiki/Submissions:Time_indexed_person_role

[7] https://www.wikidata.org

| Datasets | Dataset Size (Train/Val/Test) | Unique Entities | Relation Types | Triples per Graph |
|---|---|---|---|---|
| syn-paths | 60,000/20,000/20,000 | 49 | 3 | 3 |
| syn-types | 60,000/20,000/20,000 | 30 | 3 | 3 |
| syn-tipr | 50,000/10,000/10,000 | 130 | 5 | 5 |
| wd-movies | 38,267/15,698/15,796 | 24,093 | 3 | 2-23 |
| wd-articles | 54,163/22,922/22,915 | 60,932 | 6 | 4-212 |

Table 2: Dataset characteristics for the IntelliGraphs benchmark. Synthetic datasets (syn-*) have fixed graph structures while Wikidata-derived datasets (wd-*) exhibit variable sizes. Entity counts represent unique entities across all graphs; edge counts indicate the number of triples per individual graph.

### A.3 Methods

Here, we provide more details about the methods we used for the empirical analyses of ARK and SAIL.

#### A.3.1 Gated Recurrent Units (GRUs)

The ARK model employs a standard GRU decoder with hidden state $\mathbf{h}_t \in \mathbb{R}^d$ that evolves as:

$$\mathbf{r}_t = \sigma(\mathbf{W}_r\mathbf{x}_t + \mathbf{U}_r\mathbf{h}_{t-1} + \mathbf{b}_r) \tag{3}$$

$$\mathbf{z}_t = \sigma(\mathbf{W}_z\mathbf{x}_t + \mathbf{U}_z\mathbf{h}_{t-1} + \mathbf{b}_z) \tag{4}$$

$$\tilde{\mathbf{h}}_t = \tanh(\mathbf{W}_h\mathbf{x}_t + \mathbf{U}_h(\mathbf{r}_t \odot \mathbf{h}_{t-1}) + \mathbf{b}_h) \tag{5}$$

$$\mathbf{h}_t = (1 - \mathbf{z}_t) \odot \mathbf{h}_{t-1} + \mathbf{z}_t \odot \tilde{\mathbf{h}}_t \tag{6}$$

where $\mathbf{r}_t$ and $\mathbf{z}_t$ are reset and update gates respectively, $\mathbf{x}_t$ is the embedding of the current input token, and $\odot$ denotes element-wise multiplication. At each timestep, the hidden state is projected to vocabulary logits: $p(x_{t+1}|x_{\leq t}) = \text{softmax}(\mathbf{W}_o\mathbf{h}_t + \mathbf{b}_o)$.

#### A.3.2 Compression Length

For both ARK and SAIL, we compute the compression length to generate graphs as sequences. Since ARK is a decoder-only autoregressive model, we compute:

$$\text{Compression Length of } G = -\log_2(p_\theta(G)) = -\sum_{t=1}^{T} \log_2(p_\theta(x_t|x_{<t})) \tag{7}$$

where $x_t$ represents the $t$-th token in the linearized graph sequence $[\texttt{BOS}, h_1, r_1, t_1, ..., \texttt{EOS}]$ and $T$ is the sequence length. Each term represents the bits needed to encode the next token given the previous context.

For SAIL, the variational framework adds a latent variable $z$, resulting in an upper bound on compression length through the ELBO:

$$\text{Compression Length of } G \leq -\log_2(p(G|z)) + D_{\text{KL}}(q(z \mid G) \parallel p(z)) \tag{8}$$

$$= -\sum_{t=1}^{T} \log_2(p_\theta(x_t|x_{<t}, z)) + D_{\text{KL}} \tag{9}$$

The KL divergence term is computed as follows:

$$D_{\text{KL}}(q(z \mid G) \parallel p(z)) = \frac{1}{2}\sum_{i=1}^{d} \left(\mu_i^2 + \sigma_i^2 - 1 - \log(\sigma_i^2)\right) \cdot \log_2(e) \tag{10}$$

where $d$ is the latent dimensionality and the factor $\log_2(e)$ converts from nats to bits. The autoregressive formulation naturally handles variable-length graphs through the sequential factorization, eliminating the need for separate structure and entity terms.

This provides an upper bound on the true compression length; the VAE's ELBO is a lower bound on log-likelihood, which, when negated, becomes an upper bound on compression. The bound is particularly relevant as the autoregressive decoder must account for uncertainty in token ordering during generation.

### A.3.3 Sampling from Latent Variable, $z$

We conduct two types of generation experiments:

1. *Unconditional Generation:* We sample 10,000 random latent codes from the standard normal prior distribution $p(z) = \mathcal{N}(0, I)$ and decode them into complete graphs using beam search with beam width $k = 3$. Each decoded graph is analyzed for: (1) semantic validity according to dataset-specific constraints, (2) novelty by checking against the training and validation sets, and (3) non-emptiness to ensure the model generates meaningful structures rather than null graphs.

2. *Conditional Generation:* We evaluate the model's ability to complete partial graphs by providing incomplete sequences as prompts. For each test graph, we provide the first $n$ tokens (*e.g.*, $[\texttt{BOS}, h_1, r_1, t_1]$) and generate the remaining sequence autoregressively. We vary the conditioning length and measure: (1) the semantic validity of the completed graph and (2) the diversity of completions when sampling with different random seeds.

### A.3.4 Interpolation in Latent Space

We conduct both quantitative and qualitative analyses of the latent space structure:

1. *Quantitative Analysis:* We measure latent space smoothness through four metrics: (1) *Local Smoothness* – average Jaccard similarity between consecutive decoded graphs along random walks in latent space with step size $\epsilon = 0.1$, measuring whether small movements produce similar graphs; (2) *Global Consistency* – Jaccard similarity between each step and the anchor point, measuring drift from the starting graph; (3) *Flip Rate* – fraction of steps that produce different decoded graphs, with lower rates indicating larger basins of attraction in latent space; and (4) *Average Basin Length* – mean number of consecutive interpolation steps that decode to identical graphs, quantifying the granularity of the learned representation. For each metric, we sample multiple anchor points and random directions, taking 10-30 steps along each trajectory.

2. *Qualitative Analysis:* We visualize the latent space structure using two approaches: (1) *2D Projection* – we encode all test graphs and project their latent representations to 2D using t-SNE, coloring points by semantic attributes (genre for wd-movies) to observe clustering patterns; and (2) *Linear Interpolation* – we select pairs of semantically distinct graphs, encode them to obtain $z_1$ and $z_2$, then decode intermediate points $z_\alpha = (1 - \alpha)z_1 + \alpha z_2$ for $\alpha \in [0, 1]$ at regular intervals to examine the semantic coherence of interpolated graphs.

## A.4 Qualitative Analysis of Conditional Sampling

**Qualitative Results**    We test whether SAIL has learned meaningful latent representations that capture director-specific collaborative patterns and genre preferences, despite never being explicitly trained on individual directorial styles. Figure A.1 shows representative examples of conditional generation for director-specific movie graphs. When conditioned on "Tim Burton" as the director, the model successfully generates graphs featuring his frequent collaborators (Helena Bonham Carter, Christopher Lee) and characteristic genres (Comedy Film, Musical Film). SAIL captures Burton's tendency to work repeatedly with the same ensemble cast, demonstrating learned patterns of directorial collaboration. In contrast, the Wes Anderson generation fails to capture his distinctive style. This disparity in generation quality likely reflects differences in dataset representation; Burton's more frequent appearances and consistent casting patterns in the training

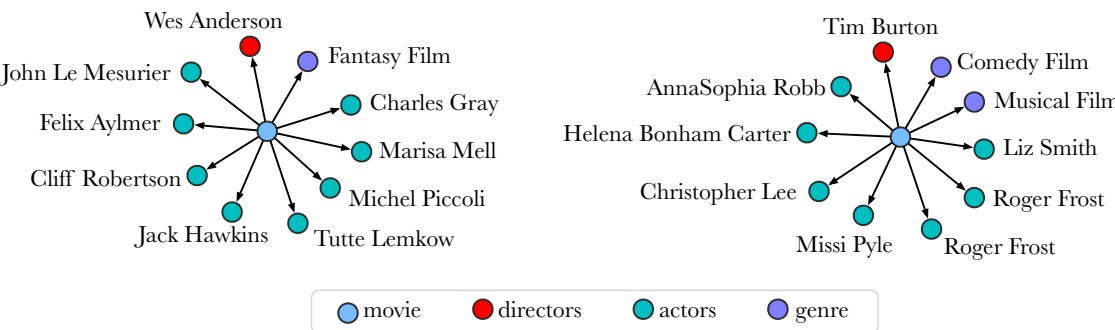

Figure A.1: Graphs generated by ARK conditioned on director entities for Wes Anderson **(left)** and (b) Tim Burton **(right)**. Node colors indicate entity types: movie (blue), directors (red), actors (green), and genres (purple).

data enabled better pattern learning, while Anderson's style may have been underrepresented. Despite these variations in director-specific accuracy, both generated graphs maintain semantic validity as movie KGs, indicating that the SAIL has learned general graph structure.

## A.5 Quantitative Analysis of Conditional Sampling

We evaluate SAIL's conditional generation quantitatively on wd-movies. The qualitative director-conditioning examples reported earlier in the paper (Tim Burton and Wes Anderson, Appendix A.4) are useful for illustration but insufficient as evidence of controlled generation: two examples cannot tell us whether the model captures conditioning identity reliably, or whether the failure on Wes Anderson is representative of a broader pattern.

**Method** For each of the top-$k$ most frequent directors in the wd-movies training set ($k = 20$), we condition SAIL on the partial graph containing only the director-of-the-movie triple and generate $N = 100$ completions per condition. We measure four properties of the conditioned generations:

1. *Director recall:* the fraction of generated graphs that include the conditioned director entity. A model that conditions correctly should yield director recall close to 1.

2. *Cast distribution match:* the Jensen-Shannon divergence between the empirical distribution of cast members in the model's conditioned generations and the empirical distribution of cast members in the training data for that director. Lower is better; a model that captures director-specific collaborative patterns should produce a low divergence.

3. *Genre distribution match:* the Jensen-Shannon divergence between the model's genre distribution under the conditioning and the training-data genre distribution for that director. Same interpretation.

4. *Validity & novelty:* semantic validity and novelty of the conditioned generations, computed under the same protocol as in the unconditional case.

**Results** SAIL achieves director recall of 1.000, indicating that the conditioned director is always retained in the generated completions. The generated graphs are also largely semantically valid and novel, with validity 98.6%, novelty 82.2%, and valid-and-novel rate 80.7%. Distributional control is stronger for genres than for casts: the mean genre JSD is 0.176, while the mean cast JSD is 0.578, suggesting that SAIL reliably follows the director condition and captures genre tendencies reasonably well, but only partially captures director-specific cast patterns.

### A.6 Ablation Study

We systematically analyze the contribution of key architectural components through two ablation experiments on the syn-paths dataset, examining both model capacity and architectural choices.

**Method** We conduct two complementary ablation studies:

1. *Architectural Hyperparameter Analysis:* We vary the number of GRU layers $n_{\text{layers}} \in \{1, 2, 3, 4, 5\}$ and model dimensions $d_{\text{model}} \in \{2, 4, 8, 16, 32, 64, 128, 256, 512\}$ while keeping other hyperparameters fixed. For each configuration, we train the model until convergence and evaluate generation by measuring the percentage of semantically valid and novel graphs. We also test the relative importance of network depth versus hidden dimensionality on generation quality.

2. *Architecture Ablation:* We systematically replace transformer components with simpler architectures to assess their contribution: (1) *MLP Encoder* – replaces the transformer encoder with a multi-layer perceptron while preserving positional encoding; (2) *GRU Decoder* – replaces the transformer decoder with a GRU-based sequential decoder; and (3) *MLP Encoder & GRU Decoder* – combines both modifications, using an MLP encoder and GRU decoder. Each variant maintains comparable parameter counts to the transformer baseline for fair comparison.

**Architectural Hyperparameter Analysis Results** In Figure A.2, the model dimension has a substantially stronger impact on generation quality than network depth. Varying the number of layers from 1 to 5 produces relatively stable performance around 45% valid & novel rate, though with high variance across configurations. In contrast, the center panel demonstrates a sharp performance threshold: models with fewer than 16 hidden units achieve near-zero validity rates, while those with $d_{\text{model}} \geq 64$ consistently achieve 70-95% validity. The right panel's scatter plot confirms this pattern across individual runs, showing clear stratification by model dimension rather than layer count (indicated by color). These findings suggest that for KG generation on syn-paths dataset, a single-layer GRU with sufficient hidden units ($\geq 64$) can match or exceed the performance of deeper networks, supporting our claim that architectural simplicity does not compromise generation quality when coupled with appropriate capacity.

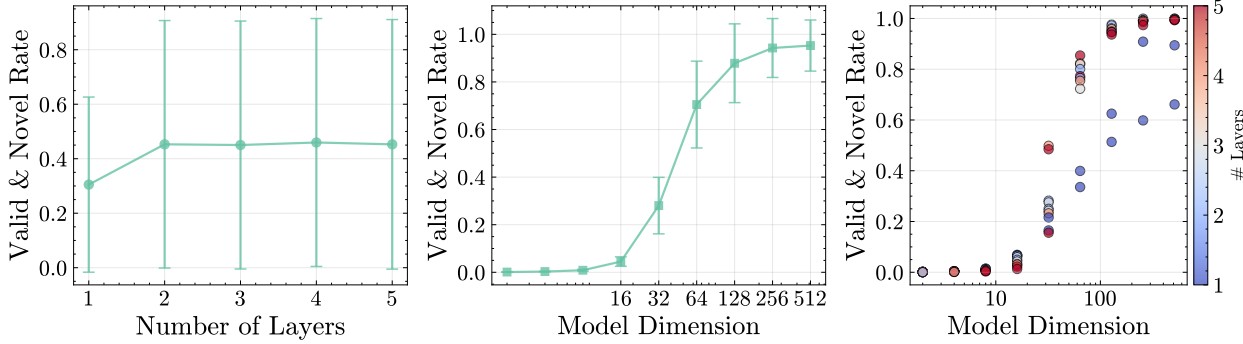

Figure A.2: Effect of architectural hyperparameters on the semantic validity and novelty. **(Left)** Valid & Novel rate as a function of the number of GRU layers, showing stable performance across depths with high variance. **(Center)** Performance variation with model dimension (hidden units), demonstrating a sharp improvement threshold around 64 dimensions, followed by consistent high performance. **(Right)** Scatter plot of individual experimental runs showing the relationship between model dimension and generation quality, with color indicating the number of layers.

**Architecture Ablation Results** To better understand the contribution of architectural choices, we compare our full transformer-based model $t$-SAIL against simplified variants: SAIL, which replaces the transformer encoder and decoder with an MLP encoder and a GRU decoder, an MLP encoder (paired with a transformer decoder), $t$-ARK, a decoder-only transformer model, and ARK, a GRU decoder-only model. These ablations allow us to isolate the effect of transformer components in both the encoder and the decoder,

and to assess whether an encoder is required for KG generation at all. In addition to generation quality and compression efficiency, we also report relative training time, as computational efficiency is often a limiting factor in scaling generative models. Table 3 demonstrates that transformer components, while improving generation quality, are not strictly necessary for effective knowledge graph modeling. Sequential decoders are consistently the most efficient: ARK trains at **0.09–0.27,$\times$** the baseline time (i.e., **3.7–11$\times$** faster) with near baseline validity across datasets, and its sequential inductive bias is competitive for decoding *e.g.*, syn-tipr (23.48 bits, on par with $t$-ARK 's 23.34) and wd-movies (**98.19** bits, best overall). Meanwhile, SAIL yields the best compression on wd-articles ( **199.55** bits), indicating that modest latent structure plus a GRU decoder can improve efficiency on complex, real-world graphs. Taken together, these results suggest that, for KG generation, a strong sequential decoder often dominates architectural choice, and the extra cost of full transformers, especially in the decoder, may be hard to justify when compute is constrained.

| Datasets | Model | % Valid Generation ↑ | % Novel Graphs ↑ | Compression (bits) ↓ | Training Time ↓ |
|---|---|---|---|---|---|
| **syn-paths** | $t$-SAIL | 99.60 | 100.00 | 27.77 | 1.00 |
| | SAIL | 92.50 | 100.00 | 28.74 | 0.21 |
| | MLP Encoder | 99.80 | 100.00 | 27.35 | 0.55 |
| | $t$-ARK | 97.39 | 100.00 | 27.57 | 0.12 |
| | ARK | 99.95 | 100.00 | 27.65 | 0.09 |
| **syn-tipr** | $t$-SAIL | 100.00 | 100.00 | 26.30 | 1.00 |
| | SAIL | 98.45 | 100.00 | 27.14 | 0.17 |
| | MLP Encoder | 99.48 | 100.00 | 26.30 | 0.20 |
| | $t$-ARK | 100.00 | 100.00 | 23.34 | 0.17 |
| | ARK | 100 | 100.00 | 23.48 | 0.09 |
| **syn-types** | $t$-SAIL | 100.00 | 100.00 | 59.61 | 1.00 |
| | SAIL | 100.00 | 100.00 | 60.58 | 0.39 |
| | MLP Encoder | 93.27 | 100.00 | 59.33 | 0.41 |
| | $t$-ARK | 87.07 | 100.00 | 59.79 | 0.18 |
| | ARK | 89.22 | 100.00 | 59.63 | 0.09 |
| **wd-movies** | $t$-SAIL | 99.83 | 100.00 | 124.50 | 1.00 |
| | SAIL | 99.47 | 100.00 | 116.84 | 0.24 |
| | MLP Encoder | 99.44 | 100.00 | 118.64 | 0.36 |
| | $t$-ARK | 98.33 | 100.00 | 114.49 | 0.23 |
| | ARK | 99.24 | 100.00 | 98.19 | 0.21 |
| **wd-articles** | $t$-SAIL | 98.00 | 96.00 | 235.24 | 1.00 |
| | SAIL | 99.13 | 100.00 | 199.55 | 0.42 |
| | MLP Encoder | 97.7 | 100.00 | 206.23 | 0.48 |
| | $t$-ARK | 95.37 | 100.00 | 224.25 | 0.33 |
| | ARK | 97.24 | 100.00 | 205.24 | 0.27 |

Table 3: Architectural ablation study comparing ARK against simplified architectures with MLP encoders and GRU decoders. We evaluate model variants across five datasets using generation quality metrics (percentage of valid and novel graphs), compression efficiency (bits required for latent representation), and computational efficiency (training time relative to $t$-SAIL baseline).

## A.7 Hidden Dimensionality vs Depth

Figure A.2 sweeps $n_{\text{layers}}$ and $d_{\text{model}}$ on syn-paths only. To check whether the same pattern shows up elsewhere, we ran the same sweep on syn-types, syn-tipr, and wd-movies, with $n_{\text{layers}} \in \{1, 2, 3, 4, 5\}$ and $d_{\text{model}} \in \{32, 64, 128, 256, 512, 1024\}$. We did not run wd-articles within the rebuttal window because of its much larger graphs and vocabulary. Table 4 reports the best test compression in bits per graph for each dataset and $d_{\text{model}}$, taken across all $n_{\text{layers}}$ values and seeds.

| Dataset | $d{=}32$ | $d{=}64$ | $d{=}128$ | $d{=}256$ | $d{=}512$ | $d{=}1024$ |
|---|---|---|---|---|---|---|
| syn-paths | 30.14 | 28.52 | 27.70 | 27.66 | **27.64** | 27.65 |
| syn-types | 65.99 | 61.47 | 60.12 | 59.75 | **59.60** | 59.63 |
| syn-tipr | 23.92 | 23.48 | **23.45** | 23.48 | 23.50 | 23.53 |
| wd-movies | 111.10 | 101.84 | **97.98** | 102.03 | 109.45 | 120.99 |

Table 4: Best test compression (bits/graph; lower is better) for ARK at each $d_{\mathrm{model}}$, across $n_{\mathrm{layers}} \in \{1..5\}$ and seeds. **Bold** marks the smallest $d_{\mathrm{model}}$ within 0.6 bits per graph of the best.

The same pattern from syn-paths shows up on the other three datasets. Hidden dimensionality has a large effect (up to 13 bits per graph), while depth has a small one: at the bolded $d_{\mathrm{model}}$, switching $n_{\mathrm{layers}}$ between 1 and 5 changes compression by less than 0.6 bits per graph. The exact $d_{\mathrm{model}}$ at which the model saturates differs across datasets ($d_{\mathrm{model}} = 64$ for syn-tipr, $d_{\mathrm{model}} = 128$ for the rest), so we no longer claim a universal threshold of $d_{\mathrm{model}} = 64$. wd-movies also shows mild overfitting at $d_{\mathrm{model}} \geq 256$, which is worth noting but does not change the headline finding.

## A.8 Conditioned Generation

Figure A.3 shows that conditioned generation is also possible for the ARK model, which allows the model to generate KGs and simultaneously enforces specific constraints. Entities or relations are fixed in place in the positions of interest, and then we decode the remaining tokens with constrained sampling (temperature/top-k/top-p). Figure A.3a shows that the novelty and validity of the generated structures remain high for all steps of the conditioning process, an indication that the model can produce triples and, consequently, graphs that are semantically correct. At the same time, as seen in Figure A.3b, the diversity of the generated graphs drops dynamically as more entities and relations are added. This makes sense as the population of probable samples narrows with each additional constraint and limits the generative freedom of the model.

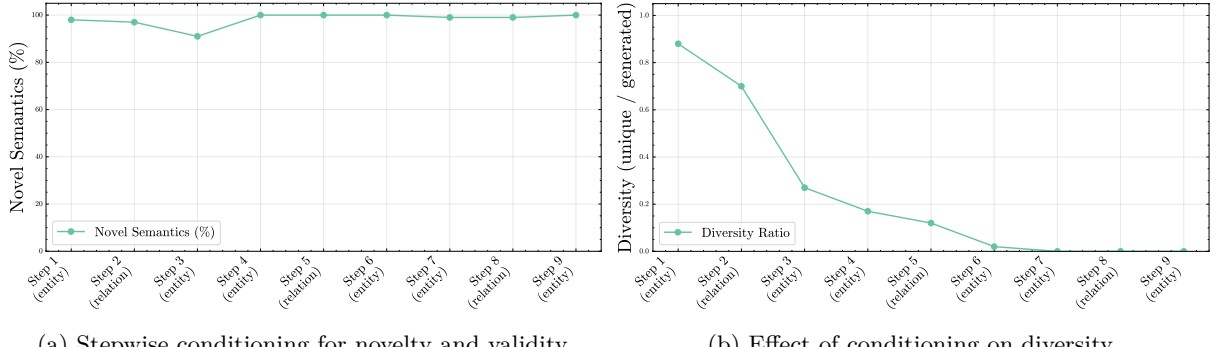

(a) Stepwise conditioning for novelty and validity.

(b) Effect of conditioning on diversity.

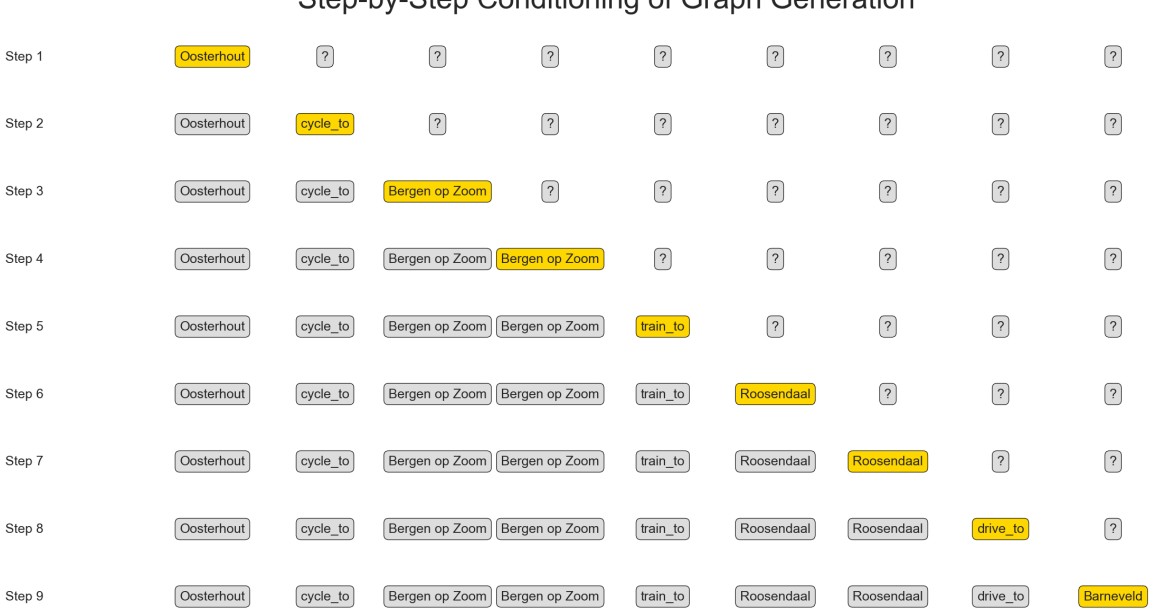

(c) Example of syn-paths conditioned generation

Figure A.3: Effect of progressive conditioning on Knowledge Graph generation for the syn-paths dataset. Subfigure (a) quantifies novelty and validity under increasing conditioning, (b) shows the corresponding reduction in sample diversity, and (c) provides an example of a conditioned generation where the model completes a partially specified graph.

### A.9 Inter-Triple Ordering Invariance

We confirm empirically that the choice of inter-triple ordering during training does not drive results. As described in Section 3, we randomize the order in which triples are presented within each graph during training to prevent leakage from dataset-specific orderings (e.g., path order in syn-paths). To verify that this design choice is benign rather than load-bearing, we trained ARK and SAIL under three different inter-triple orderings:

1. *Shuffled* (the setting reported in the main paper): the order of triples within each graph is randomized at every training pass.

2. *Alphabetical*: triples are sorted in alphabetical order starting from the head entity.

3. *Dataset-specific*: triples are presented in the order they appear in the original IntelliGraphs dataset files, which for some datasets reflects an underlying generative process (e.g., path order in syn-paths).

The results are stable across orderings: shuffled gives 99.37% semantic validity, 100.00% novelty, 99.37% valid and novel samples, and 27.67 bits/graph compression. Alphabetical order gives 96.05% semantic validity, 100.00% novelty, 96.05% valid and novel samples, and 27.51 bits/graph compression. Dataset-specific ordering gives 100.00% semantic validity, 100.00% novelty, 100.00% valid and novel samples, and 26.56 bits/graph compression. Thus, exposing the model to the original path order does not lead to a qualitatively different result: all three orderings achieve near perfect semantic validity and novelty, with only small differences in compression.

### A.10 Encoder Diagnostic for SAIL

The SAIL encoder uses a set-pooling MLP rather than a sequential encoder. As noted in Section 3, this choice was originally motivated by preliminary experiments in which a GRU encoder appeared to perform notably worse than the MLP encoder on the SAIL task, despite the GRU's stronger inductive bias for sequential data. To better understand this counter-intuitive observation, we ran a diagnostic comparison of three encoder architectures, holding the SAIL decoder and all other components fixed:

1. *MLP encoder* (the setting reported in the main paper): triples are embedded, mean-pooled across triples, then passed through dense layers.

2. *GRU encoder*: triples are embedded and read sequentially by a GRU; the final hidden state is projected to the latent parameters.

3. *Transformer encoder*: triples are embedded and processed by self-attention, then pooled to produce the latent parameters.

For each encoder, we report semantic validity and novelty as downstream generation-quality metrics, together with reconstruction loss, KL divergence, and the number of active latent dimensions as diagnostics for posterior collapse. We evaluate this comparison on representative synthetic and real-world datasets.

The diagnostic results show that the GRU encoder is not suffering from posterior collapse: it maintains non-negligible KL divergence and active latent dimensions, and its validity is comparable to the mean pooled MLP encoder. This is consistent with the main results in Table 1, where the SAIL model with the mean-pooled encoder remains competitive across datasets: it obtains 92.50% valid and novel graphs on `syn-paths`, 98.45% on `syn-tipr`, 100.00% on `syn-types`, and 99.47% on `wd-movies`.

Thus, while mean pooling is theoretically less expressive than a sequential encoder for modeling inter-triple dependencies, this limitation does not translate into a clear disadvantage in our setting. The results suggest that, for these benchmarks, the combination of triple embeddings, latent conditioning, and the autoregressive SAIL decoder is sufficient to capture the structural regularities needed for high-validity and high-novelty generation.

### A.11 Additional Comments about ARK & SAIL

**Variable Graph Length** It is desirable to learn latent graph structures of varying sizes. In natural language processing, language models utilize special tokens to indicate the end of a sequence. Following a similar approach, we model variable-length KGs by linearizing graphs into a sequence of tokens and introducing boundary tokens. We always introduce `BOS` as the initial token and terminate generation upon emitting `EOS`, while using `PAD` for mini-batching. This simple setting allows the decoder to learn *when* to stop and *how large* the generated graphs should be, ensuring that the length distribution is learned. During inference, beam search halts on `EOS`, leading to the production of graphs of different sizes without any post hoc trim. In order to avoid length bias, we randomize triple order during training. In the probabilistic variant (SAIL), the latent $\mathbf{z}$ conditions the entire sequence and this yields consistent length control across all samples, while at the same time preserving variability.

# B  Additional Tables

| Datasets | Model | % Valid Graphs ↑ | % Novel & Valid ↑ | % Novel Graphs ↑ | % Empty Graphs ↓ |
|---|---|---|---|---|---|
| **syn-paths** | uniform | 0 | 0 | 100.00 | 0 |
| | TransE | 0.25 | 0.25 | 23.45 | 76.55 |
| | DistMult | 0.69 | 0.69 | 14.59 | 85.41 |
| | ComplEx | 0.71 | 0.71 | 14.27 | 85.73 |
| | $t$-SAIL | 99.60 | 99.60 | 100.00 | 0 |
| | SAIL | 92.50 | 92.50 | 100.00 | 0 |
| | $t$-ARK | 97.39 | 97.39 | 100.00 | 0 |
| | ARK | **99.95** | **99.95** | 100.00 | 0 |
| **syn-tipr** | uniform | 0 | 0 | 100.00 | 0 |
| | TransE | 0 | 0 | 5.58 | 94.42 |
| | DistMult | 0 | 0 | 13.34 | 86.66 |
| | ComplEx | 0 | 0 | 4.95 | 96.05 |
| | $t$-SAIL | 100.00 | 100.00 | 100.00 | 0 |
| | SAIL | 98.45 | 98.45 | 100.00 | 0 |
| | $t$-ARK | 100.00 | 100.00 | 100.00 | 0 |
| | ARK | **100.00** | **100.00** | 100.00 | 0 |
| **syn-types** | uniform | 0 | 0 | 100.00 | 0 |
| | TransE | 0.21 | 0.21 | 15.44 | 84.56 |
| | DistMult | 0.13 | 0.13 | 12.46 | 87.53 |
| | ComplEx | 0.07 | 0.07 | 10.25 | 89.75 |
| | $t$-SAIL | 100.00 | **100.00** | **100.00** | 0 |
| | SAIL | 100.00 | 100.00 | 100.00 | 0 |
| | $t$-ARK | 87.07 | 87.07 | 100.00 | 0 |
| | ARK | 89.22 | 89.22 | 100.00 | 0 |
| **wd-movies** | uniform | 0 | 0 | 100.00 | 0 |
| | TransE | 0 | 0 | 14.61 | 85.39 |
| | DistMult | 0 | 0 | 12.93 | 87.07 |
| | ComplEx | 0 | 0 | 1.87 | 98.13 |
| | $t$-SAIL | **99.83** | **99.9** | **100** | 0 |
| | SAIL | 99.47 | 99.47 | 100.00 | 0 |
| | $t$-ARK | 98.33 | 98.33 | 100.00 | 0 |
| | ARK | 99.24 | 99.24 | 100.00 | 0 |
| **wd-articles** | uniform | 0 | 0 | 100.00 | 0 |
| | TransE | 0 | 0 | 4.58 | 95.42 |
| | DistMult | 0 | 0 | 0 | 100.00 |
| | ComplEx | 0 | 0 | 2.46 | 97.54 |
| | $t$-SAIL | 98.00 | 98.00 | 100.00 | 0 |
| | SAIL | **99.13** | **99.13** | 100.00 | 0 |
| | $t$-ARK | 95.37 | 95.37 | 100.00 | 0 |
| | ARK | 97.24 | 97.24 | 99.99 | 0 |

Table 5: Semantic validity of the graphs generated. We sample graphs and check the novelty of the sampled graphs by comparing them against the training and validation sets. The best performing models for each dataset are **bolded**. Baseline results are from the IntelliGraphs paper (Thanapalasingam et al., 2023).

| Datasets | Models | Compression Length (bits) | | | |
|---|---|---|---|---|---|
| | | $G$ | $S$ | $E$ | $D_{KL}$ |
| **syn-paths** | uniform | 30.49 | 12.80 | 17.69 | - |
| | TransE | 49.89 | 16.19 | 33.69 | - |
| | ComplEx | 54.39 | 20.71 | 33.69 | - |
| | DistMult | 48.58 | 14.90 | 33.69 | - |
| | $t$-SAIL | 27.77 | - | 14.47 | 13.30 |
| | SAIL | 28.74 | - | 18.41 | 10.33 |
| | $t$-ARK | **27.57** | - | - | - |
| | ARK | 27.65 | - | - | - |
| **syn-tipr** | uniform | 61.61 | 29.14 | 32.47 | - |
| | TransE | 69.51 | 28.70 | 40.81 | - |
| | ComplEx | 63.96 | 23.15 | 40.81 | - |
| | DistMult | 67.51 | 26.70 | 40.81 | - |
| | $t$-SAIL | 26.30 | - | 11.13 | 15.17 |
| | SAIL | 27.14 | - | 9.90 | 17.24 |
| | $t$-ARK | **23.34** | - | - | - |
| | ARK | 23.48 | - | - | - |
| **syn-types** | uniform | **36.02** | 16.84 | 19.18 | - |
| | TransE | 48.26 | 19.05 | 29.21 | - |
| | ComplEx | 47.69 | 18.48 | 29.21 | - |
| | DistMult | 47.46 | 18.24 | 29.21 | - |
| | $t$-SAIL | 59.61 | - | 59.46 | 0.15 |
| | SAIL | 60.58 | - | 60.37 | 0.21 |
| | $t$-ARK | 59.79 | - | - | - |
| | ARK | 59.63 | - | - | - |
| **wd-movies** | uniform | 171.60 | 53.86 | 117.74 | - |
| | TransE | 208.60 | 51.39 | 157.21 | - |
| | ComplEx | 202.68 | 45.46 | 157.21 | - |
| | DistMult | 208.50 | 51.29 | 157.21 | - |
| | $t$-SAIL | 124.50 | - | 92.66 | 31.84 |
| | SAIL | 116.84 | - | 100.10 | 16.74 |
| | $t$-ARK | 114.49 | - | - | - |
| | ARK | **98.19** | - | - | - |
| **wd-articles** | uniform | 693.80 | 295.60 | 398.20 | - |
| | TransE | 910.65 | 280.67 | 629.98 | - |
| | ComplEx | 887.30 | 257.33 | 629.98 | - |
| | DistMult | 901.91 | 271.94 | 629.98 | - |
| | $t$-SAIL | 235.24 | - | 225.60 | 9.64 |
| | SAIL | **199.55** | - | 186.38 | 13.17 |
| | $t$-ARK | 224.25 | - | - | - |
| | ARK | 205.24 | - | - | - |

Table 6: We measure the compression quality for compressing graphs $G$. $D_{KL}$ is only available for the VAE because it relies on the variational approximation, which is unique to this model. For the VAE, we compute an upper bound on the compression length (in bits). Probabilistic baseline (uniform, TransE, ComplEx, DistMult) results are from Thanapalasingam et al. (2023).

# C  Additional Figures

## C.1  Architectural Details

Figure C.1 shows the architectural details of the $t$-SAIL model. Also Figure A.1 shows an example for conditioned generation.

| Dataset | Model | Local Smoothness ↑ | Global Consistency ↑ | Flip Rate ↓ | Avg Basin Length ↑ |
|---|---|---|---|---|---|
| **syn-paths** | $t$-SAIL | 0.75 | 0.36 | 0.20 | 4.54 |
| | SAIL | 0.74 | 0.14 | 0.33 | 2.87 |
| **syn-tipr** | $t$-SAIL | 0.99 | 0.98 | 0.09 | 8.61 |
| | SAIL | 0.93 | 0.69 | 0.10 | 8.03 |
| **syn-types** | $t$-SAIL | 0.82 | 0.60 | 0.12 | 6.80 |
| | SAIL | 0.92 | 0.73 | 0.20 | 4.47 |
| **wd-movies** | $t$-SAIL | 0.87 | 0.58 | 0.15 | 5.70 |
| | SAIL | 0.84 | 0.49 | 0.40 | 2.93 |
| **wd-articles** | $t$-SAIL | 0.81 | 0.55 | 0.14 | 5.37 |
| | SAIL | 0.82 | 0.57 | 0.17 | 3.46 |

Table 7: Latent space smoothness metrics for $t$-SAIL and SAIL with $\epsilon = 0.1$. Higher local/global smoothness indicates more continuous transitions. Lower flip rates suggest larger regions mapping to identical graphs.

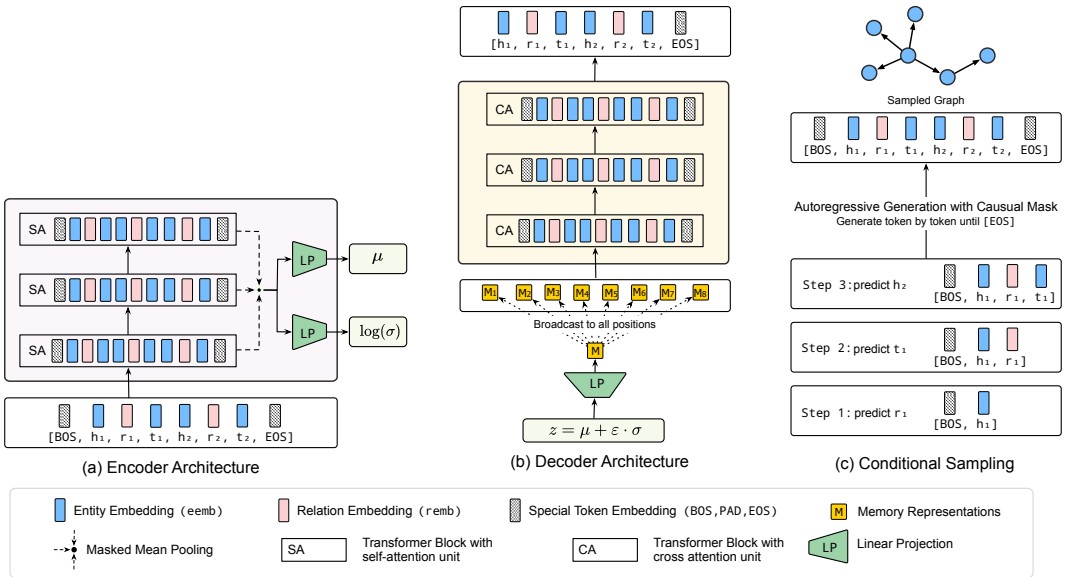

Figure C.1: $t$-SAIL has three main components: (a) an *Encoder* that processes linearized Knowledge Graph triple sequences $[\texttt{BOS}, h_1, r_1, t_1, h_2, r_2, t_2, \ldots, \texttt{EOS}]$ through self-attention (SA) blocks to produce latent distribution parameters $(\mu, \log \sigma)$, (b) a *Decoder* that uses cross-attention (CA) to condition on the sampled latent code $z$ and autoregressively generates token sequences with causal masking, and (c) *Conditional Sampling* that demonstrates the step-by-step autoregressive generation process, predicting one token at a time until the $\texttt{[EOS]}$ token is produced or the maximum sequence length is reached. The model uses a unified vocabulary embedding matrix spanning special tokens ($\texttt{[BOS], [PAD], [EOS]}$), entities (shown in blue), and relations (shown in pink), enabling sequential generation of Knowledge Graphs from learned latent representations.

