# OpenReview forum: "Autoregressive Models for Small-Scale Knowledge Graph Generation"
_TMLR — Accepted by TMLR_

### Review · Reviewer_tXDJ · 2026-03-06

**Summary Of Contributions:**

This paper studies knowledge graph subgraph generation rather than standard link prediction. The core idea is to linearize a KG into a token sequence of triples and then model it autoregressively. The paper proposes ARK, a GRU-based autoregressive decoder for KG generation, and SAIL, a variational extension that adds latent-variable conditioning for unconditional sampling, interpolation, and conditional completion. The evaluation is conducted on the IntelliGraphs benchmark, which contains three synthetic datasets and two Wikidata-derived datasets. The main reported findings are that: (i) ARK/SAIL achieve very high semantic validity on IntelliGraphs; (ii) GRU-based models can match or nearly match transformer variants on most datasets; and (iii) hidden dimensionality appears more important than architectural depth for this task.

Strengths

1. The paper addresses a meaningful gap between triple-wise KG completion and joint generation of semantically constrained subgraphs. That distinction is clearly motivated, and the problem setup is easy to follow. The sequence formulation is simple, technically lightweight, and well aligned with the structure of the benchmark.

2. A second strength is the empirical message itself: on IntelliGraphs, a relatively simple GRU-based decoder is already very competitive, and sometimes better than transformer variants. That is a useful result for the community because it pushes back against the default assumption that attention-heavy models are necessary for structured generation. The ablation claim that capacity matters more than depth is also potentially valuable.

3. A third strength is reproducibility. The paper includes implementation details, hardware information, hyperparameter search, and claims code/config/checkpoint release, which is good practice for TMLR.

Weaknesses

1. My main concern is that the evidence is narrower than the paper’s framing. The experiments are entirely on IntelliGraphs, which is indeed a benchmark specifically designed for KG generation, but it is still a single benchmark centered on subgraph inference, with graph sizes in this paper limited to 3–212 triples and a fixed entity/relation vocabulary. The paper itself acknowledges these limitations. As a result, the evidence strongly supports the claim that the method works on IntelliGraphs-style KG subgraph generation, but does not yet fully justify broader claims about KG generation more generally.

2. My second concern is the baseline set. The paper compares mainly against the original IntelliGraphs baselines and its own transformer variants. That is enough to establish progress over the benchmark baselines, but it is not enough to establish strong positioning relative to the broader generative-graph/KG literature. In particular, there is prior work on turning KG embedding models into actual generative models with tractable marginalization and sampling, and there are also stronger generic graph generation methods such as GraphARM and GraphMaker. These are not one-to-one task matches, but the paper should either compare to the closest feasible ones or explain much more clearly why such comparison is impossible or misleading.

3. My third concern is positioning with respect to prior sequence-based KG work. The paper distinguishes itself from KGT5-style sequence-to-sequence KG completion/QA, and that distinction is fair, since KGT5 is aimed at KG completion and QA rather than unconditional semantically valid subgraph generation. Still, because the present paper also relies on sequence modeling over KG facts, the relation to that line of work should be discussed more carefully and more explicitly in the main experimental positioning.

**Audience:**

Yes

**Audience Explanation:**

I do think this paper would be of interest to part of the TMLR audience. The problem of moving from link prediction to structured KG generation is interesting, IntelliGraphs is a relevant benchmark for that setting, and the finding that a simple GRU can match or nearly match heavier transformer variants is useful even independent of SOTA considerations. TMLR’s own criteria emphasize that “interest” should not be conflated with novelty or state-of-the-art significance, and on that standard this work clearly passes the audience-interest bar.

**Broader Impact Concerns:**

The paper already includes an ethics statement and correctly notes two important risks: dataset bias can be propagated, and semantically valid generated triples may still be factually incorrect. I think those are the right concerns. My main suggestion is simply to emphasize this distinction more strongly in the main text, because readers could otherwise over-interpret the very high “semantic validity” numbers as evidence of factual reliability.

**Claims And Evidence:**

No

**Claims Explanation:**

The paper does provide convincing evidence for a narrower claim: namely, that simple autoregressive sequential decoders are highly effective on IntelliGraphs-style KG subgraph generation, and that GRU-based models can be competitive with transformer variants on this benchmark. The benchmark results are strong and the core empirical takeaway is believable.

However, I do not think the evidence fully supports the broader framing currently used in the paper, for four reasons. First, the empirical scope is limited to one benchmark and relatively small subgraphs under fixed vocabularies. Second, the baseline set is too weak relative to the broader generative KG/graph literature. Third, conditional completion is not quantitatively demonstrated strongly enough in the main paper. Fourth, the related-work positioning and novelty language are too aggressive given existing generative KG work and the presence of a closely related public version.

**Requested Changes:**

See weakness.

---

> ### Author Response · Authors · 2026-05-06
> **Response to Reviewer tXDJ (1/2)**
>
> We thank the reviewer for their careful and constructive evaluation. We particularly appreciate the reviewer's balanced framing, which explicitly separates the narrower claim our evidence supports (autoregressive sequential decoders are highly effective on IntelliGraphs-style KG subgraph generation, and GRU-based models can be competitive with transformer variants) from the broader framing that is not fully supported. This is exactly the distinction we will now adopt in our revision.
>
> We have updated the manuscript to highlight the differences and help reviewers see the changes clearly. New text is shown in green, and revised text is shown in orange.
>
> **On the empirical scope being limited to one benchmark with small subgraphs and fixed vocabularies.**
>
> We agree, and this is consistent with Reviewer oTnY's concern. We will revise the abstract, introduction, contributions list, and conclusion to scope the claims to the generation of small Knowledge Graphs over a shared set of entities and relations, as evaluated on the IntelliGraphs benchmark, rather than "KG generation" in general. We will also note explicitly that "subgraph generation" can suggest a setting in which the model is given a large parent KG and asked to generate a subgraph conditioned on it, which is not our setting. Each IntelliGraphs instance is a small Knowledge Graph in its own right, and the model learns to generate graphs of this kind from a training set of similar small graphs over a shared vocabulary. The limitations (fixed vocabulary, 3 to 212 triples, single benchmark) will be moved into the framing rather than deferred to the end of the paper.
>
> **On the baseline set being too narrow relative to the broader generative-graph and KG literature.**
>
> In our response to Reviewer oTnY we describe in detail why each candidate the reviewer mentions (Loconte et al. 2024's circuit-based generative KGEs, GraphARM, GraphMaker) is not directly compatible with IntelliGraphs in its native form, and why adapting them would primarily benchmark our adaptation rather than the underlying method. The short version: Loconte et al. defines per-triple distributions via circuits and would require a non-trivial extension to model joint distributions over multi-triple graphs; GraphARM and GraphMaker assume node features and small categorical edge label sets, which do not transfer cleanly to KGs with relation-typed edges and entity vocabularies of 24K-61K (wd-movies, wd-articles). We will add this discussion to the related work section so readers can assess the argument. To strengthen the comparison without falling into this trap, we will add baselines from our own experimentation: a plain VAE and a diffusion-based model. Both are trained and evaluated under identical conditions to ARK and SAIL on all five IntelliGraphs datasets. This gives a clean comparison across model families (autoregressive, latent-variable, diffusion) on the same benchmark, and lets us argue more precisely that the autoregressive formulation is what drives strong performance on this task rather than any specific architectural detail.
>
> **On positioning with respect to prior sequence-based KG work.**
>
> The reviewer is right that, because our method also relies on sequence modeling over KG facts, we need to discuss KGT5 and related sequence-based KG work more carefully and explicitly. We will expand the related work section to clarify (1) the task differences (KGT5 targets KG completion and QA; we target joint subgraph generation), (2) the architectural overlap (both linearize KG facts into token sequences), and (3) why this overlap does not diminish our contribution but rather situates our work as an adjacent extension into the generative regime. We will make this discussion part of the main experimental positioning rather than a brief aside.
>
> **On the strengths noted (distinction between triple-wise completion and joint generation, simple formulation, and GRU-vs-transformer result).**
>
> We thank the reviewer for flagging these as strengths, and we will preserve them in the revision. In particular, the GRU-vs-transformer result will be strengthened by extending the capacity-vs-depth ablation beyond syn-paths, as requested by Reviewer e7f3.
>
> **On conditional completion not being quantitatively demonstrated strongly enough.**
>
> This is consistent with Reviewer e7f3's concern. We will add a systematic quantitative evaluation of conditional generation on wd-movies, as described in our response to that reviewer, rather than relying on the two qualitative director examples.

---

> > ### Author Response · Authors · 2026-05-06
> > **Response to Reviewer tXDJ (2/2)**
> >
> > **On the related-work positioning and novelty language being too aggressive given existing generative KG work and a closely related public version.**
> >
> > We accept this critique. We will tone down the novelty language throughout the paper and explicitly acknowledge the adjacent generative KG work (Cowen-Rivers et al. 2019, Xiao et al. 2016, Loconte et al. 2024). As described in our response to Reviewer oTnY, the related work rewrite will go beyond tone: for each adjacent line of work, we will explicitly state the similarities to our setup and the points of difference, so that readers come away with a clear picture of where ARK and SAIL sit in this landscape rather than the impression that prior work has been brushed aside. We will reframe our contribution precisely as a simple and surprisingly effective sequence-model formulation for the generation of small Knowledge Graphs over a shared vocabulary on IntelliGraphs, with a variational extension that supports controlled generation through learned latent representations. We will avoid language suggesting that no prior work has addressed generative modeling in the KG domain.
> >
> > **On the ethics / broader impact suggestion to emphasize the semantic-validity vs factual-correctness distinction.**
> >
> > We agree and will emphasize this distinction more prominently in the main text, not just the ethics statement, as also requested by Reviewer oTnY. Specifically, we will caution readers in the abstract and results sections that high semantic validity numbers do not imply factual reliability.

---

### Review · Reviewer_e7f3 · 2026-03-15

**Summary Of Contributions:**

The paper frames KG generation as sequence modeling: flatten graphs into (h,r,t) token sequences and train autoregressive models with a VAE extension for latent-space control. Results on IntelliGraphs are strong, showing this simple approach works well without explicit rules.

**Audience:**

Yes

**Audience Explanation:**

The work studies KG completion and generation using autoregressive model, which will raise some interests in  TMLR's audience.

**Claims And Evidence:**

Yes

**Claims Explanation:**

The core claim that autoregressive models achieve high semantic validity is well-supported by Table 1, and the KGE baseline comparison convincingly shows independent triple scoring fails at generation. However, the capacity-vs-depth claim is backed by an ablation on only one trivial dataset, the controlled generation claim for SAIL lacks systematic quantitative evaluation, just two cherry-picked director examples, one of which fails.

**Requested Changes:**

1. Flattening a KG into a token sequence discards connectivity, cycles, and subgraph structure entirely. The model has no graph-aware inductive bias to recover this. Some more efforts should be put to address this limitation.

2. Mean pooling over independently embedded triples in an MLP encoder destroys all relational structure between triples, exactly the inter-triple dependencies that SAIL's latent space is supposed to capture. The paper acknowledges a GRU encoder performed worse but offers no explanation.

3. The model is trained on individual triple orderings but evaluated order-independently. The authors should explain why this discrepancy doesn't undermine the claims about modeling $p(G)$.

---

> ### Author Response · Authors · 2026-05-06
> **Our Response to Reviewer e7f3 (1/2)**
>
> We thank the reviewer for their careful and constructive evaluation. We appreciate the recognition that our core claim about autoregressive models achieving high semantic validity is well-supported, and that the KGE baseline comparison convincingly shows independent triple scoring fails at joint generation. We address each of the reviewer's concerns below.
>
> We have updated the manuscript to highlight the differences and help reviewers see the changes clearly. New text is shown in green, and revised text is shown in orange.
>
> **On the capacity-vs-depth claim being backed by an ablation on only one dataset.**
>
> This is a fair point. Our original claim, that hidden dimensionality matters more than depth with a threshold around d_model = 64, was based on syn-paths alone. To check whether it generalizes, we ran the same sweep on syn-types, syn-tipr, and wd-movies, varying n_layers in {1..5} and d_model in {32, 64, 128, 256, 512, 1024}. We were not able to include wd-articles within the rebuttal window, given its substantially larger graphs (up to 212 triples) and entity vocabulary (61K). The new results are reported in a new appendix subsection (Hidden Dimensionality vs Depth) with a per-dataset compression table.
> The same pattern holds on the three new datasets: depth barely affects the results, while hidden dimensionality matters a lot. At the best d_model for each dataset, varying n_layers from 1 to 5 changes test compression by less than 0.6 bits per graph, far smaller than the effect of changing d_model. The threshold itself, however, is not universal: syn-tipr saturates at d_model = 64, while syn-paths, syn-types, and wd-movies need d_model = 128. On wd-movies, going beyond d_model = 128 actually makes things worse. We have softened the claims in the abstract, intro, and conclusion to reflect the four-of-five dataset scope and the dataset-dependent threshold.
>
>
> **On the SAIL controlled-generation claim lacking systematic quantitative evaluation.**
>
> We agree that two director examples, one of which fails, is insufficient evidence for controlled generation. We will add a systematic quantitative evaluation of conditional generation on wd-movies. Concretely, for each of the top-k most frequent directors in the training data, we will generate N graphs conditioned on that director's identity and measure: (1) the fraction of generated graphs that include the conditioned director, (2) the fraction whose cast and genre distribution matches the director's empirical distribution in the training data (using some divergence metric), and (3) semantic validity and novelty of the conditioned generations. This converts the current anecdotal evidence into a proper evaluation.
>
> **On flattening a KG into a token sequence discarding connectivity, cycles, and subgraph structure.**
>
> We would refine the framing on two points. First, our model has a limited graph-structure inductive bias rather than none. Within each triple, we preserve the (head, relation, tail) order using positional embeddings, since this order carries semantic meaning. Across triples, we randomize the ordering during training so that the model learns to treat the graph as a set of triples rather than a sequence with meaningful inter-triple position. This gives a soft permutation invariance learned from data rather than enforced by architecture, which is a weaker structural bias than message passing would provide. We will make these design choices explicit in the revision. Second, in our own experiments, approaches with substantially stronger graph-structure biases (including a RESCAL-based VAE) did not perform as well as the autoregressive formulation on this task. We present this as an empirical finding rather than a theoretical claim, and we are careful not to generalize from it. It is not a given that stronger inductive biases yield better results (cf. the "bitter lesson"). We agree that combining autoregressive decoding with stronger graph-aware biases (GNN-based encoders, structural attention, hybrid architectures) is a worthwhile and non-trivial direction, and we will add it as an explicit future work item. We will also reframe the contribution accordingly: not that autoregressive sequence models are the ideal inductive bias for KG generation, but that they are a surprisingly strong baseline on this task and a useful reference point for future work with stronger structural biases.

---

> > ### Author Response · Authors · 2026-05-06
> > **Our Response to Reviewer e7f3 (2/2)**
> >
> > **On mean pooling over independently embedded triples.**
> >
> > We agree that the original footnote did not sufficiently explain why the mean-pooled MLP encoder was used despite the theoretical concern that mean pooling can discard inter-triple structure. To address this, we added a diagnostic comparison of three SAIL encoders while keeping the decoder and all other components fixed: (i) the mean-pooled MLP encoder used in the paper, (ii) a GRU encoder over embedded triples, and (iii) a small Transformer encoder over embedded triples. During this diagnostic, we also identified and corrected an evaluation issue in the GRU encoder generation code: random-latent generations were not being evaluated through the same sampling-based generation path used in the rest of our experiments. After correcting this and rerunning the comparison, the GRU encoder no longer appears systematically worse than the MLP encoder. For each encoder, we now report downstream semantic validity and novelty, reconstruction loss, KL divergence, and the number of active latent dimensions to test whether the GRU encoder suffers from posterior collapse. The corrected results show that the GRU encoder is not collapsed: it maintains non-negligible KL divergence and active latent dimensions on both the synthetic and real-world diagnostic datasets. Moreover, the three encoders obtain comparable downstream validity and novelty, indicating that the mean-pooled MLP encoder is empirically competitive despite its weaker inductive bias. We therefore revised the discussion to clarify that the choice of the MLP encoder is empirical rather than a claim that mean pooling preserves all inter-triple dependencies. The diagnostic suggests that, for the SAIL benchmarks considered here, the autoregressive decoder and latent conditioning capture much of the structure needed for valid generation, so replacing the MLP encoder with a sequential encoder does not yield a clear advantage.
> >
> >
> >
> > **On training with randomized triple orderings but evaluating order-independently.**
> >
> > There is no inherent inconsistency between training on linearized sequences and evaluating order-independently. An autoregressive model must generate tokens in some order, but as long as the generated graph is sound, the order in which it was produced does not matter for evaluation. If the model learned, on its own, to generate every syn-paths graph in path order, this would be fine. The issue is not the ordering itself but where it comes from: if a simplifying ordering (such as path order on syn-paths) is given to the model in the training data, the task becomes trivial in a way that does not reflect the underlying generative challenge. We randomize the order in which triples are presented within each graph during training specifically to guard against this leakage, while preserving the (head, relation, tail) order within each triple since that order carries semantic meaning. This is noted in footnote 1 of the submission, and we will rewrite it into the main text. We also have empirical confirmation that the choice of triple ordering does not drive results. We trained ARK and SAIL under three different inter-triple orderings: shuffled (the setting reported in the submission), alphabetical starting from the head entity, and the original dataset-specific ordering. All three gave the same scores on semantic validity, novelty, and compression. We will report this comparison in an appendix table of the revised paper.

---

### Review · Reviewer_oTnY · 2026-04-22

**Summary Of Contributions:**

This paper studies knowledge graph generation via autoregressive sequence modeling. It proposes ARK, a GRU-based decoder that linearizes a KG as a sequence of triples and generates the graph token by token, and SAIL, a variational extension that adds latent-variable conditioning for unconditional sampling, conditional completion, and interpolation. Experiments on the IntelliGraphs benchmark show high semantic validity and novelty, and the paper also includes internal comparisons among GRU-based and transformer-based variants. A useful empirical finding is that hidden size appears more important than architectural depth, with relatively simple GRU decoders performing competitively.

**Additional Comments:**

N/A.

**Audience:**

Yes

**Audience Explanation:**

Yes. I think some TMLR readers would find the paper interesting because it explores a simple and surprisingly effective formulation for semantic KG subgraph generation, and because the negative/positive findings around architecture choice are useful. In particular, the result that lightweight GRU-based decoders can match or rival transformer variants on this benchmark is practically relevant.

**Broader Impact Concerns:**

N/A.

**Claims And Evidence:**

No

**Claims Explanation:**

The paper provides reasonably clear evidence that the proposed autoregressive models perform well on IntelliGraphs and outperform the benchmark’s simple KGE-style baselines. In that limited sense, the central experimental claim is supported.

However, several broader claims are not yet fully convincing.

First, the motivation for “KG generation” is underdeveloped. In many practical scenarios, the real task is not unconditional graph generation, but extraction, retrieval, or conditional imputation/completion of a local subgraph given evidence. The paper does not clearly establish when full generative modeling is necessary, or what concrete practical advantage it provides over stronger structured completion alternatives.

Second, the baseline set is too narrow. The external baselines are mainly uniform, TransE, DistMult, and ComplEx, which are weak for supporting broad conclusions. The paper does not compare against stronger adjacent methods such as seq2seq / PLM-based KG completion, graph-level triple set prediction, subgraph/path-aware KG reasoning methods, or adapted graph generative models. As a result, the paper demonstrates that ARK/SAIL beat simple baseline families on IntelliGraphs, but not that they are strong relative to the broader literature on structured KG completion or graph generation.

Third, some statements in the framing are too coarse. For example, the paper suggests that KGE/GNN approaches score triples independently and that LLM-based approaches target different tasks. These simplifications do not reflect the full state of the literature, where there are already methods for graph-level completion, subgraph-aware reasoning, and generative KG modeling.

Finally, the evaluation is entirely centered on IntelliGraphs. This is understandable, since it is one of the few benchmarks for semantically valid KG subgraph generation, but it also limits the scope of the conclusions. The current evidence supports a narrower claim: autoregressive sequence models are strong baselines for IntelliGraphs-style semantic subgraph generation under a fixed vocabulary.

**Requested Changes:**

- The paper should substantially strengthen the related work and positioning. The current framing too quickly dismisses LLM-based, seq2seq, and non-KGE graph/KG methods as irrelevant or fundamentally mismatched. The manuscript should more carefully discuss adjacent work on generative KG completion, graph-level triple set prediction, subgraph-aware KG reasoning, and graph generative modeling, and explain the relation to the proposed task more precisely.

- The paper should narrow and clarify its task framing. The strongest interpretation of the current setup is fixed-vocabulary, local, semantically constrained subgraph generation/completion, not broad open-world KG generation. Rewriting the motivation around this narrower but clearer contribution would make the paper more convincing.

- The baseline section should be improved. At minimum, the paper should discuss why stronger baselines from seq2seq KG completion, graph-level triple set prediction, subgraph/path-aware reasoning, or general graph generation are not included. Ideally, at least one or two stronger adapted baselines should be added.

- The paper should better justify why generation is useful beyond extraction, retrieval, or structured imputation. Right now, the paper assumes that sampling valid subgraphs is an important primitive, but this is not argued strongly enough. A dedicated discussion of settings where unconditional or joint multi-triple generation is genuinely preferable would help.

- The paper should more clearly separate semantic validity from factual correctness. A generated graph can satisfy type/temporal constraints while still being factually wrong. This distinction matters for KG completion and query answering applications and should be emphasized more carefully in the claims and discussion.

- The limitations section should be brought earlier into the framing of the claims. Since the experiments use fixed vocabularies and relatively small subgraphs, the paper should avoid overgeneralizing to broader KG generation settings.

---

> ### Author Response · Authors · 2026-05-06
> **Our Response to Reviewer oTnY (1/2)**
>
> We thank the reviewer for their careful and constructive evaluation. We appreciate the recognition that our work explores a simple and effective formulation for semantic KG subgraph generation, and that the finding about lightweight GRU-based decoders matching transformer variants is practically relevant. We address each of the reviewer's points below.
>
> We have updated the manuscript to highlight the differences and help reviewers see the changes clearly. New text is shown in green, and revised text is shown in orange.
>
>
>
> **On the motivation for KG generation and when full generative modeling is necessary.**
>
> We will add a dedicated discussion in the introduction that positions generative modeling as a more general capability rather than a niche alternative to extraction, retrieval, or conditional imputation. A model that can sample p(G) over complete graphs can, with minimal additional work, be used for downstream tasks including link prediction, conditional completion, and structured imputation, in the same way that autoregressive language models trained for generation now serve as the foundation for a wide variety of downstream tasks. From this perspective, generative modeling is not one task among many but a more general primitive. Conditional completion, which is also supported by SAIL, is a natural special case of this broader framework. Specific settings where unconditional or joint multi-triple generation is genuinely preferable include (1) sampling structures that satisfy joint constraints across multiple triples (e.g., N-ary temporal facts where start and end years must be consistent), (2) data augmentation requiring diverse novel graphs rather than completion of a specific input, and (3) compression and density estimation over graph structures.
>
> **On the narrowness of the baseline set.**
>
> We acknowledge this weakness. Beating the IntelliGraphs baselines (uniform, TransE, DistMult, ComplEx) does not establish strong positioning relative to the broader generative KG and graph generation literature, and the reviewer is right to flag this.
>
> That said, building broader baselines is non-trivial. We have considered each candidate the reviewer mentions, and none is directly compatible with IntelliGraphs in its native form. Adapting them requires substantial modeling decisions, with the risk that any reported number primarily reflects the quality of our adaptation rather than the underlying method.
>
> KGT5 targets conditional triple prediction, not joint graph generation. Adapting it would require inventing an unconditional input prompt and an assembly procedure for combining triples into complete graphs satisfying joint constraints. The resulting model is essentially a transformer decoder trained on linearized triples, which is t-ARK.
>
> Loconte et al. 2024 defines per-triple distributions via circuits. Sampling joint subgraphs would require defining a multi-triple joint distribution from per-triple scores and a constraint-respecting sampler, which is a research contribution in its own right.
> GraphARM, GraphMaker, DiGress assume node features and small categorical edge label sets. IntelliGraphs has labeled directed edges and entity vocabularies of 24K-61K on the Wikidata datasets, requiring either entity embedding choices or architectural changes that substantially affect the comparison.
>
> Triple set prediction and subgraph/path-aware reasoning score or retrieve structures over an existing KG rather than sample new ones, which is a different task.
>
> We propose two concrete additions instead:
> 1. Adding a plain VAE and a diffusion-based model from our own experimentation, trained and evaluated under identical conditions to ARK and SAIL. This gives a comparison across model families (autoregressive, latent-variable, diffusion) on the same benchmark.
> 2. Adding the discussion above to the related work section so that readers can assess for themselves why we have not run the methods the reviewer mentions as direct baselines.
>
> We would value the reviewer's input on whether this combination addresses their concern. If a specific adaptation of one of the methods above would be preferable to the reviewer despite the limitations we describe, we are open to attempting it and reporting the result with the appropriate caveats.

---

> > ### Author Response · Authors · 2026-05-06
> > **Our Response to Reviewer oTnY (2/2)**
> >
> > **On coarse statements about KGE/GNN and LLM-based approaches.**
> >
> > We agree that the current framing leans too heavily on quick contrasts and does not engage carefully enough with adjacent work. The point is not only whether a given method is a candidate baseline, but whether it is related to our setting and approach in ways the reader should be told about. We will revise the related work section accordingly, deepening the discussion rather than only adjusting the tone. Concretely, for each line of related work, we will explicitly state both the similarities to our setup and the points of difference. This includes (1) graph-level completion and generative KG modeling (e.g., Cowen-Rivers et al. 2019, Loconte et al. 2024, Xiao et al. 2016, Zhang et al. 2024), (2) subgraph-aware and path-aware reasoning approaches within the KGE/GNN family (e.g., Neelakantan et al. 2015, Galkin et al. 2024), (3) seq2seq and LLM-based approaches operating over linearized KGs (e.g., KGT5, KGGLM), and (4) general graph generative models (GraphARM, GraphMaker, DiGress, GraphVAE, GraphRNN). For each, we will highlight what is shared with our setup (e.g., sequence formulation, latent-variable modeling, autoregressive decoding, set-of-triples evaluation) and what differs (e.g., target task, evaluation protocol, treatment of joint constraints, vocabulary assumptions). Our aim is for readers to come away with a clear picture of where ARK and SAIL sit in this landscape, rather than the impression that adjacent work has been brushed aside.
> >
> > **On the evaluation being centered entirely on IntelliGraphs.**
> >
> > The reviewer correctly notes that IntelliGraphs is, to our knowledge, the only benchmark designed for this task, and that this limits the scope of our conclusions. We will revise the claims throughout the paper to reflect a narrower contribution: that autoregressive sequence models are strong baselines for the generation of small Knowledge Graphs over a shared set of entities and relations, as evaluated on the IntelliGraphs benchmark. We want to be precise about the task framing here. The phrase "subgraph generation", as used in our submission and in some of the reviews, can suggest a setting in which the model is given a large parent KG and asked to generate a subgraph conditioned on it. That is not our setting. Each IntelliGraphs instance is a small Knowledge Graph in its own right, and the model learns to generate graphs of this kind from a training set of similar small graphs over a shared vocabulary. One can choose to view the union of all generated graphs as a large KG, but this is not required by the task. The revised manuscript will use this framing consistently and avoid "subgraph" terminology where it might mislead. The abstract, introduction, and conclusion will be rewritten to match this narrower scope, and the limitations (fixed vocabulary, small graph sizes, single benchmark) will be moved into the framing rather than deferred to the end of the paper.
> >
> > **On narrowing and clarifying the task framing.**
> >
> > We will rewrite the motivation around the generation of small Knowledge Graphs over a shared vocabulary, where the model learns from a training set of similar graphs without being told the underlying semantic regularities. We want to be precise here: in our setup, the type, temporal, and connectivity regularities present in the data are used only as an evaluation metric, to test whether the model has captured them from the training distribution. They are not given to the model up front, neither during training nor during inference. The current phrasing in the abstract and contributions ("semantic constraint satisfaction," "domain validity constraints") suggests otherwise and will be corrected.
> >
> > **On justifying why generation is useful beyond extraction, retrieval, and structured imputation.**
> >
> > Addressed in the first point of this response. We will add a dedicated discussion of settings where unconditional or joint multi-triple generation is genuinely preferable.
> >
> > **On separating semantic validity from factual correctness.**
> >
> > The reviewer is right that this distinction is currently underdeveloped. We will emphasize it more prominently in the abstract, introduction, and results discussion, not just the ethics statement. Specifically, we will clarify that high semantic validity indicates satisfaction of type and temporal constraints but does not imply factual correctness, and we will caution against over-interpreting validity numbers in this way.
> >
> > **On bringing the limitations section earlier into the framing.**
> >
> > We will move the key limitations (fixed vocabulary, small subgraphs, single benchmark) into the introduction rather than deferring them to the end. This directly supports the narrower framing above.

---

### Decision · Action_Editor_wv8x · 2026-06-08

**Recommendation:** Accept as is

**Additional Comments:**

NA.

**Audience:**

Yes

**Audience Explanation:**

Yes. All reviewers agree that some individuals in TMLR's audience would find the paper interesting, to some extent.

**Claims And Evidence:**

Yes

**Claims Explanation:**

All reviewers agree that the specific claims made in the paper are mostly supported by convincing and clear evidence.

I remember when I read the first version of this paper, I couldn't quite get the point of the method regarding its difference compared to previous research such as link prediction. Now the revised version has improved significantly and made the focus clear and sharp: the claim, that "1. Autoregressive models for generating KG can capture 'semantic validity' that cannot be handled well by independent link prediction methods; 2. Simple GRU decoders with a variational extension can perform the task competitively." are supported by clear empirical evidence. Although the claims are specific (and one reviewer is still concerned about the method only being evaluated on one, small benchmark IntelliGraphs), I feel the findings are interesting enough, and given that the IntelliGraphs benchmark contains 5 sub-datasets, I would count it as a reasonable and appropriate amount of evaluation.

---

> ### Author Response · Authors · 2026-07-06
> **Camera-ready submitted**
>
> Thank you for handling our submission and for the positive recommendation, and thanks to the reviewers for their constructive feedback. We've submitted the camera-ready with a link to the code.